# TAMING GANS WITH LOOKAHEAD–MINMAX

**Tatjana Chavdarova**[*]
EPFL

**Mattéo Pagliardini**[*]
EPFL

**Sebastian U. Stich**
EPFL

**François Fleuret**
University of Geneva

**Martin Jaggi**
EPFL

## ABSTRACT

Generative Adversarial Networks are notoriously challenging to train. The underlying minmax optimization is highly susceptible to the variance of the stochastic gradient and the rotational component of the associated game vector field. To tackle these challenges, we propose the Lookahead algorithm for minmax optimization, originally developed for single objective minimization only. The backtracking step of our Lookahead–minmax naturally handles the rotational game dynamics, a property which was identified to be key for enabling gradient ascent descent methods to converge on challenging examples often analyzed in the literature. Moreover, it implicitly handles high variance without using large mini-batches, known to be essential for reaching state of the art performance. Experimental results on MNIST, SVHN, CIFAR-10, and ImageNet demonstrate a clear advantage of combining Lookahead–minmax with Adam or extragradient, in terms of performance and improved stability, for negligible memory and computational cost. Using 30-fold fewer parameters and 16-fold smaller minibatches we outperform the reported performance of the class-dependent BigGAN on CIFAR-10 by obtaining FID of 12.19 *without* using the class labels, bringing state-of-the-art GAN training within reach of common computational resources. Our source code is available:
`https://github.com/Chavdarova/LAGAN-Lookahead_Minimax`.

## 1 INTRODUCTION

Gradient-based methods are the workhorse of machine learning. These methods optimize the parameters of a model with respect to a single objective $f \colon \mathcal{X} \to \mathbb{R}$. However, an increasing interest for multi-objective optimization arises in various domains—such as mathematics, economics, multi-agent reinforcement learning (Omidshafiei et al., 2017)—where several agents aim at optimizing their own cost function $f_i \colon \mathcal{X}_1 \times \cdots \times \mathcal{X}_{\mathcal{N}} \to \mathbb{R}$ simultaneously.

A particularly successful class of algorithms of this kind are the Generative Adversarial Networks (Goodfellow et al., 2014, (GANs)), which consist of two players referred to as a generator and a discriminator. GANs were originally formulated as minmax optimization $f \colon \mathcal{X} \times \mathcal{Y} \to \mathbb{R}$ (Von Neumann & Morgenstern, 1944), where the generator and the discriminator aim at minimizing and maximizing the same value function, see § 2. A natural generalization of gradient descent for minmax problems is the gradient descent ascent algorithm (GDA), which alternates between a gradient descent step for the min-player and a gradient ascent step for the max-player. This minmax training aims at finding a Nash equilibrium where no player has the incentive of changing its parameters.

Despite the impressive quality of the samples generated by the GANs—relative to classical maximum likelihood-based generative models—these models remain *notoriously difficult to train*. In particular, poor performance (sometimes manifesting as "mode collapse"), brittle dependency on hyperparameters, or divergence are often reported. Consequently, obtaining state-of-the-art performance was shown to require large computational resources (Brock et al., 2019), making well-performing models unavailable for common computational budgets.

---

[*]Equal contributions. Correspondence to `firstname.lastname@epfl.ch`.

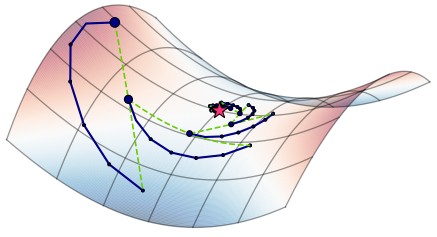

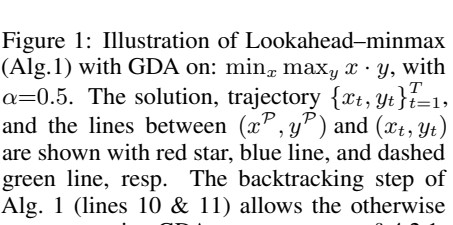

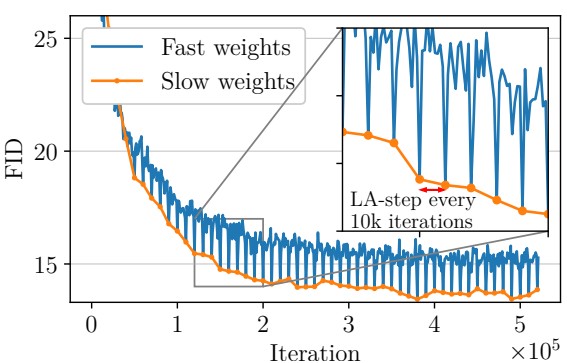

Figure 1: Illustration of Lookahead–minmax (Alg.1) with GDA on: $\min_x \max_y x \cdot y$, with $\alpha{=}0.5$. The solution, trajectory $\{x_t, y_t\}_{t=1}^T$, and the lines between $(x^{\mathcal{P}}, y^{\mathcal{P}})$ and $(x_t, y_t)$ are shown with red star, blue line, and dashed green line, resp. The backtracking step of Alg. 1 (lines 10 & 11) allows the otherwise non-converging GDA to converge, see § 4.2.1.

Figure 2: FID ($\downarrow$) of Lookahead–minmax on $32{\times}32$ ImageNet (§ 5), see Fig. 12 for IS. We observe *significant* improvements after each LA-step, what empirically confirms the existence of rotations and thus the intuition of the behaviour of LA-minmax on games, as well as the relevance of the bilinear game for real world applications–Fig.1.

It was empirically shown that: (i) GANs often converge to a locally stable stationary point that is *not* a differential Nash equilibrium (Berard et al., 2020); (ii) increased batch size improves GAN performances (Brock et al., 2019) in contrast to minimization (Defazio & Bottou, 2019; Shallue et al., 2018). A principal reason is attributed to the *rotations* arising due to the adversarial component of the associated vector field of the gradient of the two player's parameters (Mescheder et al., 2018; Balduzzi et al., 2018), which are atypical for minimization. More precisely, the Jacobian of the associated vector field (see def. in § 2) can be decomposed into a symmetric and antisymmetric component (Balduzzi et al., 2018), which behave as a *potential* (Monderer & Shapley, 1996) and a *Hamiltonian* game, resp. Games are often combination of the two, making this general case harder to solve.

In the context of single objective minimization, Zhang et al. (2019) recently proposed the *Lookahead* algorithm, which intuitively uses an update direction by "looking ahead" at the sequence of parameters that change with higher variance due to the "stochasticity" of the gradient estimates–called *fast* weights–generated by an inner optimizer. Lookahead was shown to improve the stability during training and to reduce the variance of the so called *slow* weights.

**Contributions.** Our contributions can be summarized as follows:

- We propose Lookahead–minmax for optimizing minmax problems, that applies extrapolation in the *joint* parameter space (see Alg 1), so as to account for the rotational component of the associated game vector field (defined in § 2).

- In the context of: (i) *single objective minimization*: by building on insights of Wang et al. (2020), who argue that Lookahead can be interpreted as an instance of local SGD, we derive improved convergence guarantees for the Lookahead algorithm; (ii) *two-player games*: we elaborate why Lookahead–minmax suppresses the rotational part in a simple bilinear game, and prove its convergence for a given converging base-optimizer; in § 3 and 4, resp.

- We motivate the use of Lookahead–minmax for games by considering the extensively studied toy bilinear example (Goodfellow, 2016) and show that: (i) the use of lookahead allows for convergence of the otherwise diverging GDA on the classical bilinear game in full-batch setting (see § 4.2.1), as well as (ii) it yields good performance on challenging stochastic variants of this game, despite the high variance (see § 4.2.2).

- We empirically benchmark Lookahead–minmax on GANs on four standard datasets— MNIST, CIFAR-10, SVHN and ImageNet—*on two different models* (DCGAN & ResNet), with standard optimization methods for GANs, GDA and Extragradient, called LA–AltGAN and LA–ExtraGradient, resp. We *consistently* observe both stability and performance improvements at a negligible additional cost that does not require additional forward and backward passes, see § 5.

## 2 BACKGROUND

**GAN formulation.** Given the data distribution $p_d$, the generator is a mapping $G \colon z \mapsto x$, where $z$ is sampled from a known distribution $z \sim p_z$ and ideally $x \sim p_d$. The discriminator $D \colon x \mapsto D(x) \in [0, 1]$ is a binary classifier whose output represents a conditional probability estimate that an $x$ sampled from a balanced mixture of real data from $p_d$ and $G$-generated data is actually real. The optimization of a GAN is formulated as a differentiable two-player game where the generator $G$ with parameters $\boldsymbol{\theta}$, and the discriminator $D$ with parameters $\boldsymbol{\varphi}$, aim at minimizing their own cost function $\mathcal{L}^{\boldsymbol{\theta}}$ and $\mathcal{L}^{\boldsymbol{\varphi}}$, respectively, as follows:

$$\boldsymbol{\theta}^\star \in \arg\min_{\boldsymbol{\theta} \in \Theta} \mathcal{L}^{\boldsymbol{\theta}}(\boldsymbol{\theta}, \boldsymbol{\varphi}^\star) \qquad \text{and} \qquad \boldsymbol{\varphi}^\star \in \arg\min_{\boldsymbol{\varphi} \in \Phi} \mathcal{L}^{\boldsymbol{\varphi}}(\boldsymbol{\theta}^\star, \boldsymbol{\varphi}). \tag{2P-G}$$

When $\mathcal{L}^{\boldsymbol{\varphi}} = -\mathcal{L}^{\boldsymbol{\theta}}$ the game is called a *zero-sum* and equation 2P-G is a minmax problem.

**Minmax optimization methods.** As GDA does *not* converge for some simple convex-concave game, Korpelevich (1976) proposed the *extragradient* method, where a "prediction" step is performed to obtain an extrapolated point $(\boldsymbol{\theta}_{t+\frac{1}{2}}, \boldsymbol{\varphi}_{t+\frac{1}{2}})$ using GDA, and the gradients at the *extrapolated* point are then applied to the current iterate $(\boldsymbol{\theta}_t, \boldsymbol{\varphi}_t)$ as follows:

$$\text{Extrapolation:} \begin{cases} \boldsymbol{\theta}_{t+\frac{1}{2}} = \boldsymbol{\theta}_t - \eta \nabla_{\boldsymbol{\theta}} \mathcal{L}^{\boldsymbol{\theta}}(\boldsymbol{\theta}_t, \boldsymbol{\varphi}_t) \\ \boldsymbol{\varphi}_{t+\frac{1}{2}} = \boldsymbol{\varphi}_t - \eta \nabla_{\boldsymbol{\varphi}} \mathcal{L}^{\boldsymbol{\varphi}}(\boldsymbol{\theta}_t, \boldsymbol{\varphi}_t) \end{cases} \text{Update:} \begin{cases} \boldsymbol{\theta}_{t+1} = \boldsymbol{\theta}_t - \eta \nabla_{\boldsymbol{\theta}} \mathcal{L}^{\boldsymbol{\theta}}(\boldsymbol{\theta}_{t+\frac{1}{2}}, \boldsymbol{\varphi}_{t+\frac{1}{2}}) \\ \boldsymbol{\varphi}_{t+1} = \boldsymbol{\varphi}_t - \eta \nabla_{\boldsymbol{\varphi}} \mathcal{L}^{\boldsymbol{\varphi}}(\boldsymbol{\theta}_{t+\frac{1}{2}}, \boldsymbol{\varphi}_{t+\frac{1}{2}}) \end{cases} \tag{EG}$$

where $\eta$ denotes the step size. In the context of zero-sum games, the extragradient method converges for any *convex-concave* function $\mathcal{L}$ and any closed convex sets $\Theta$ and $\Phi$ (Facchinei & Pang, 2003).

**The joint vector field.** Mescheder et al. (2017) and Balduzzi et al. (2018) argue that the vector field obtained by concatenating the gradients of the two players gives more insights of the dynamics than studying the loss surface. The joint vector field (JVF) and the Jacobian of JVF are defined as:

$$v(\boldsymbol{\theta}, \boldsymbol{\varphi}) = \begin{pmatrix} \nabla_{\boldsymbol{\theta}} \mathcal{L}^{\boldsymbol{\theta}}(\boldsymbol{\theta}, \boldsymbol{\varphi}) \\ \nabla_{\boldsymbol{\varphi}} \mathcal{L}^{\boldsymbol{\varphi}}(\boldsymbol{\theta}, \boldsymbol{\varphi}) \end{pmatrix}, \text{ and } v'(\boldsymbol{\theta}, \boldsymbol{\varphi}) = \begin{pmatrix} \nabla_{\boldsymbol{\theta}}^2 \mathcal{L}^{\boldsymbol{\theta}}(\boldsymbol{\theta}, \boldsymbol{\varphi}) & \nabla_{\boldsymbol{\varphi}} \nabla_{\boldsymbol{\theta}} \mathcal{L}^{\boldsymbol{\theta}}(\boldsymbol{\theta}, \boldsymbol{\varphi}) \\ \nabla_{\boldsymbol{\theta}} \nabla_{\boldsymbol{\varphi}} \mathcal{L}^{\boldsymbol{\varphi}}(\boldsymbol{\theta}, \boldsymbol{\varphi}) & \nabla_{\boldsymbol{\varphi}}^2 \mathcal{L}^{\boldsymbol{\varphi}}(\boldsymbol{\theta}, \boldsymbol{\varphi}) \end{pmatrix}, \text{ resp.} \tag{JVF}$$

**Rotational component of the game vector field.** Berard et al. (2020) show empirically that GANs converge to a *locally stable stationary point* (Verhulst, 1990, LSSP) that is not a differential Nash equilibrium–defined as a point where the norm of the Jacobian is zero and where the Hessian of both the players are *definite positive*, see § C. LSSP is defined as a point $(\boldsymbol{\theta}^\star, \boldsymbol{\varphi}^\star)$ where:

$$v(\boldsymbol{\theta}^\star, \boldsymbol{\varphi}^\star) = 0, \quad \text{and} \quad \mathcal{R}(\lambda) > 0, \forall \lambda \in Sp(v'(\boldsymbol{\theta}^\star, \boldsymbol{\varphi}^\star)), \tag{LSSP}$$

where $Sp(\cdot)$ denotes the spectrum of $v'(\cdot)$ and $\mathcal{R}(\cdot)$ the real part. In summary, (i) if all the eigenvalues of $v'(\boldsymbol{\theta}_t, \boldsymbol{\varphi}_t)$ have positive real part the point $(\boldsymbol{\theta}_t, \boldsymbol{\varphi}_t)$ is LSSP, and (ii) if the eigenvalues of $v'(\boldsymbol{\theta}_t, \boldsymbol{\varphi}_t)$ have imaginary part, the dynamics of the game exhibit rotations.

**Impact of noise due to the stochastic gradient estimates on games.** Chavdarova et al. (2019) point out that relative to minimization, noise impedes more the game optimization, and show that there exists a class of zero-sum games for which the *stochastic* extragradient method *diverges*. Intuitively, bounded noise of the stochastic gradient hurts the convergence as with higher probability the noisy gradient points in a direction that makes the algorithm to diverge from the equilibrium, due to the properties of $v'(\cdot)$ (see Fig.1, Chavdarova et al., 2019).

## 3 LOOKAHEAD FOR SINGLE OBJECTIVE

In the context of single objective minimization, Zhang et al. (2019) recently proposed the Lookahead algorithm where at every step $t$: (i) a copy of the current iterate $\tilde{\boldsymbol{\omega}}_t$ is made: $\tilde{\boldsymbol{\omega}}_t \leftarrow \boldsymbol{\omega}_t$, (ii) $\tilde{\boldsymbol{\omega}}_t$ is then updated for $k \geq 1$ times yielding $\tilde{\boldsymbol{\omega}}_{t+k}$, and finally (iii) the actual update $\boldsymbol{\omega}_{t+1}$ is obtained as a *point that lies on a line between the two iterates*: the current $\boldsymbol{\omega}_t$ and the predicted one $\tilde{\boldsymbol{\omega}}_{t+k}$:

$$\boldsymbol{\omega}_{t+1} \leftarrow \boldsymbol{\omega}_t + \alpha(\tilde{\boldsymbol{\omega}}_{t+k} - \boldsymbol{\omega}_t), \qquad \text{where } \alpha \in [0, 1]. \tag{LA}$$

**Algorithm 1** General Lookahead–Minmax pseudocode.

1: **Input:** Stopping time $T$, learning rates $\eta_{\boldsymbol{\theta}}, \eta_{\boldsymbol{\varphi}}$, initial weights $\boldsymbol{\theta}_0, \boldsymbol{\varphi}_0$, lookahead $k$ and $\alpha$, losses $\mathcal{L}^{\boldsymbol{\theta}}$, $\mathcal{L}^{\boldsymbol{\varphi}}, \boldsymbol{d}^{\boldsymbol{\varphi}}_{(\cdot)}, \boldsymbol{d}^{\boldsymbol{\theta}}_{(\cdot)}$ base optimizer updates defined in § 4, $p_d$ and $p_z$ real and noise–data distributions, resp.
2: $\tilde{\boldsymbol{\theta}}_{0,0}, \tilde{\boldsymbol{\varphi}}_{0,0} \leftarrow \boldsymbol{\theta}_0, \boldsymbol{\varphi}_0$
3: **for** $t \in 0, \ldots, T-1$ **do**
4:     **for** $i \in 0, \ldots, k-1$ **do**
5:         **Sample** $\boldsymbol{x}, \boldsymbol{z} \sim p_d, p_z$
6:         $\tilde{\boldsymbol{\varphi}}_{t,i+1} = \tilde{\boldsymbol{\varphi}}_{t,i} - \eta_{\boldsymbol{\varphi}} \boldsymbol{d}^{\boldsymbol{\varphi}}_{t,i}(\tilde{\boldsymbol{\theta}}_{t,i}, \tilde{\boldsymbol{\varphi}}_{t,i}, \boldsymbol{x}, \boldsymbol{z})$
7:         **Sample** $\boldsymbol{z} \sim p_z$
8:         $\tilde{\boldsymbol{\theta}}_{t,i+1} = \tilde{\boldsymbol{\theta}}_{t,i+1} - \eta_{\boldsymbol{\theta}} \boldsymbol{d}^{\boldsymbol{\theta}}_{t,i}(\tilde{\boldsymbol{\theta}}_{t,i}, \tilde{\boldsymbol{\varphi}}_{t,i}, \boldsymbol{z})$
9:     **end for**
10:    $\boldsymbol{\varphi}_{t+1} = \boldsymbol{\varphi}_t + \alpha_{\boldsymbol{\varphi}}(\tilde{\boldsymbol{\varphi}}_{t,k} - \boldsymbol{\varphi}_t)$
11:    $\boldsymbol{\theta}_{t+1} = \boldsymbol{\theta}_t + \alpha_{\boldsymbol{\theta}}(\tilde{\boldsymbol{\theta}}_{t,k} - \boldsymbol{\theta}_t)$
12:    $\tilde{\boldsymbol{\theta}}_{t+1,0}, \tilde{\boldsymbol{\varphi}}_{t+1,0} \leftarrow \boldsymbol{\theta}_{t+1}, \boldsymbol{\varphi}_{t+1}$
13: **end for**
14: **Output:** $\boldsymbol{\theta}_T, \boldsymbol{\varphi}_T$

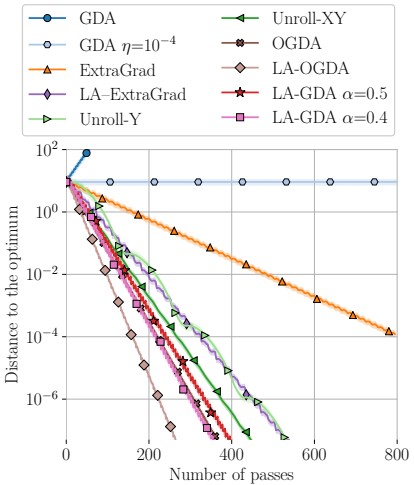

Figure 3: Distance to the optimum of equation SB–G using different *full–batch* methods, averaged over 5 runs. Unless otherwise specified, the learning rate is $\eta = 0.3$. See § 4.2.1.

Lookahead uses two additional hyperparameters: (i) $k$–the number of steps used to obtain the prediction $\tilde{\omega}_{t+k}$, as well as (ii) $\alpha$– that controls how large a step we make towards the predicted iterate $\tilde{\omega}$: the larger the closest, and when $\alpha = 1$ equation LA is equivalent to regular optimization (has no impact). Besides the extra hyperparameters, LA was shown to help the used optimizer to be more resilient to the choice of its hyperparameters, achieve faster convergence across different tasks, as well as to reduce the variance of the gradient estimates (Zhang et al., 2019).

**Theoretical Analysis.** Zhang et al. (2019) study LA on quadratic functions and Wang et al. (2020) recently provided an analysis for general smooth non-convex functions. One of their main observations is that LA can be viewed as an instance of local SGD (or parallel SGD, Stich, 2019; Koloskova et al., 2020; Woodworth et al., 2020b) which allows us to further tighten prior results.

**Theorem 1.** *Let $f \colon \mathbb{R}^d \to \mathbb{R}$ be L-smooth (possibly non-convex) and assume access to unbiased stochastic gradients $\sigma^2$-bounded variance. Then the LA optimizer with hyperparemeters $(k, \alpha)$ converges to a stationary point $\mathbb{E}\|\nabla f(\omega_{\mathrm{out}})\|^2 \leq \varepsilon$ (for the proof refer to Appendix A), after at most*

$$\mathcal{O}\left(\frac{\sigma^2}{\varepsilon^2} + \frac{1}{\varepsilon} + \frac{1-\alpha}{\alpha}\left(\frac{\sigma\sqrt{k-1}}{\varepsilon^{3/2}} + \frac{k}{\varepsilon}\right)\right)$$

*iterations. Here $\omega_{\mathrm{out}}$ denotes uniformly at random chosen iterate of LA.*

**Remark 1.** *When in addition $f$ is also quadratic, the complexity estimate improves to $\mathcal{O}\left(\frac{\sigma^2}{\varepsilon^2} + \frac{1}{\varepsilon}\right)$.* The asymptotically most significant term, $\mathcal{O}\left(\frac{\sigma^2}{\varepsilon^2}\right)$, matches with the corresponding term in the SGD convergence rate for all choices of $\alpha \in (0, 1]$, and when $\alpha \to 1$, the same convergence guarantees as for SGD can be attained. When $\sigma^2 = 0$ the rate improves to $\mathcal{O}\left(\frac{1}{\varepsilon}\right)$, in contrast to $\mathcal{O}\left(\frac{1}{\varepsilon^2}\right)$ in (Wang et al., 2020).[1] For small values of $\alpha$, the worst-case complexity estimates of LA can in general be $k$ times worse than for SGD (except for quadratic functions, where the rates match). Deriving tighter analyses for LA that corroborate the observed practical advantages is still an open problem.

## 4 LOOKAHEAD FOR MINMAX OBJECTIVES

We now study the LA optimizer in the context of minmax optimization. We start with an illustrative example, considering the bilinear game: $\min_{\boldsymbol{\theta}} \max_{\boldsymbol{\varphi}} \boldsymbol{\theta}^{\top} I \boldsymbol{\varphi}$ (see Fig. 1).
**Observation 1: Gradient Descent Ascent always diverges.** The iterates of GDA are:

$$\boldsymbol{\omega}_{t+1} \triangleq \begin{bmatrix} \boldsymbol{\theta}_{t+1} \\ \boldsymbol{\varphi}_{t+1} \end{bmatrix} = \begin{bmatrix} \boldsymbol{\theta}_t \\ \boldsymbol{\varphi}_t \end{bmatrix} + \eta \cdot \begin{bmatrix} -\boldsymbol{\varphi}_t \\ \boldsymbol{\theta}_t \end{bmatrix}.$$

---

[1] Our results rely on optimally tuned stepsize $\gamma$ for every choice of $\alpha$. Wang et al. (2020) state a slightly different observation but keep the stepize $\gamma$ fixed for different choices of $\alpha$.

The norm of the iterates $\|\boldsymbol{\omega}_{t+1}\|^2 = (1 + \eta^2)\|\boldsymbol{\omega}_t\|^2$ increases for any stepsize $\eta > 0$, hence *GDA diverges* for all choices of $\eta$.

In this bilinear game, taking a point on a line between two points on a cyclic trajectory would reduce the distance to the optimum—as illustrated in Fig. 1—what motivates extending the Lookahead algorithm to games, while using equation LA in joint parameter space $\boldsymbol{\omega} \triangleq (\boldsymbol{\theta}, \boldsymbol{\varphi})$.

**Observation 2: Lookahead *can* converge.** The iterates of LA with $k = 2$ steps of GDA are:

$$\boldsymbol{\omega}_{t+1} \triangleq \begin{bmatrix} \boldsymbol{\theta}_{t+1} \\ \boldsymbol{\varphi}_{t+1} \end{bmatrix} = \begin{bmatrix} \boldsymbol{\theta}_t \\ \boldsymbol{\varphi}_t \end{bmatrix} + \alpha\eta \cdot \begin{bmatrix} -2\boldsymbol{\varphi}_t - \eta\boldsymbol{\theta}_t \\ 2\boldsymbol{\theta}_t - \eta\boldsymbol{\varphi}_t \end{bmatrix}$$

We can again compute the norm: $\|\boldsymbol{\omega}_{t+1}\|^2 = \left((1 - \eta^2\alpha)^2 + 4\eta^2\alpha^2\right)\|\boldsymbol{\omega}_t\|^2$. For the choice $\alpha \in (0, \frac{2}{\eta^2+4})$ the norm strictly decreases, hence the algorithm converges linearly.

## 4.1 THE GENERAL LOOKAHEAD-MINMAX ALGORITHM AND ITS CONVERGENCE

Alg. 1 summarizes the proposed Lookahead–minmax algorithm which for clarity does *not* cover *different update ratios* for the two players–see Alg. 3 for this case. At every step $t = 1, \ldots, T$, we first compute k-step "predicted" iterates $(\tilde{\boldsymbol{\theta}}_{t,k}, \tilde{\boldsymbol{\varphi}}_{t,k})$—also called *fast* weights—using an inner "base" optimizer whose updates for the two players are denoted with $\boldsymbol{d}^\varphi$ and $\boldsymbol{d}^\theta$. For GDA and EG we have:

$$\text{GDA:} \begin{cases} \boldsymbol{d}_{t,i}^\varphi \triangleq \nabla_{\tilde{\boldsymbol{\varphi}}} \mathcal{L}^\varphi(\tilde{\boldsymbol{\theta}}_{t,i}, \tilde{\boldsymbol{\varphi}}_{t,i}, \boldsymbol{x}, \boldsymbol{z}) \\ \boldsymbol{d}_{t,i}^\theta \triangleq \nabla_{\tilde{\boldsymbol{\theta}}} \mathcal{L}^\theta(\tilde{\boldsymbol{\theta}}_{t,i}, \tilde{\boldsymbol{\varphi}}_{t,i}, \boldsymbol{z}) \end{cases} \quad \text{EG:} \begin{cases} \boldsymbol{d}_{t,i}^\varphi \triangleq \nabla_{\tilde{\boldsymbol{\varphi}}} \mathcal{L}^\varphi(\tilde{\boldsymbol{\theta}}_{t,i+\frac{1}{2}}, \tilde{\boldsymbol{\varphi}}_{t,i+\frac{1}{2}}, \boldsymbol{x}, \boldsymbol{z}) \\ \boldsymbol{d}_{t,i}^\theta \triangleq \nabla_{\tilde{\boldsymbol{\theta}}} \mathcal{L}^\theta(\tilde{\boldsymbol{\theta}}_{t,i+\frac{1}{2}}, \tilde{\boldsymbol{\varphi}}_{t,i+\frac{1}{2}}, \boldsymbol{z}) \end{cases},$$

where for the latter, $\boldsymbol{\varphi}_{t,i+\frac{1}{2}}, \boldsymbol{\theta}_{t,i+\frac{1}{2}}$ are computed using equation EG. The above updates can be combined with *Adam* (Kingma & Ba, 2015), see § G for detailed descriptions of these algorithms. The so called *slow* weights $\boldsymbol{\theta}_{t+1}, \boldsymbol{\varphi}_{t+1}$ are then obtained as a point on a line between $(\boldsymbol{\theta}_t, \boldsymbol{\varphi}_t)$ and $(\tilde{\boldsymbol{\theta}}_{t,k}, \tilde{\boldsymbol{\varphi}}_{t,k})$—*i.e.* equation LA, see lines 10 & 11 (herein called *backtracking* or LA–step). When combined with GDA, Alg. 1 can be *equivalently* re-written as Alg. 3, and note that it differs from using Lookahead separately per player–see § G.2. Moreover, Alg. 1 can be applied several times using *nested* LA–steps with (different) $k$, see Alg. 6.

**Local convergence.** We analyze the game dynamics as a discrete dynamical system as in (Wang et al., 2020; Gidel et al., 2019b). Let $F(\boldsymbol{\omega})$ denote the operator that an optimization method applies to the iterates $\boldsymbol{\omega} = (\boldsymbol{\theta}, \boldsymbol{\varphi})$, *i.e.* $\boldsymbol{\omega}_t = F \circ \cdots \circ F(\boldsymbol{\omega}_0)$. For example, for GDA we have $F(\boldsymbol{\omega}) = \boldsymbol{\omega} - \eta v(\boldsymbol{\omega})$. A point $\boldsymbol{\omega}^\star$ is called *fixed point* if $F(\boldsymbol{\omega}^\star) = \boldsymbol{\omega}^\star$, and it is stable if the spectral radius $\rho(\nabla F(\boldsymbol{\omega}^\star)) \leq 1$, where for GDA for example $\nabla F(\boldsymbol{\omega}^\star) = I - \eta v'(\boldsymbol{\omega}^\star)$. In the following, we show that Lookahead-minmax combined with a base optimizer which converges to a stable fixed point, also converges to the same fixed point. On the other hand, its convergence when combined with a diverging base-optimizer depends on the choice of its hyper-parameters. Developing a practical algorithm for finding optimal set of hyper-parameters for realistic setup is out of the scope of this work, and in § 5 we show empirically that Alg. 1 outperforms its base-optimizer for all tested hyperparameters.

**Theorem 2.** *(Proposition 4.4.1 Bertsekas, 1999) If the spectral radius $\rho(\nabla F(\boldsymbol{\omega}^\star)) < 1$, the $\boldsymbol{\omega}^\star$ is a point of attraction and for $\boldsymbol{\omega}_0$ in a sufficiently small neighborhood of $\boldsymbol{\omega}^\star$, the distance of $\boldsymbol{\omega}_t$ to the stationary point $\boldsymbol{\omega}^\star$ converges at a linear rate.*

The above theorem guarantees local linear convergence to a fixed point for particular class of operators. The following general theorem (proof in § B) further shows that for any such base optimizer (of same operator class) which converges linearly to the solution, Lookahead–Minmax converges too, what includes Proximal Point, EG and OGDA, but does not give precise rates—left for future work.

**Theorem 3.** *(Convergence of Lookahead–Minmax, given converging base optimizer) If the spectral radius of the Jacobian of the operator of the base optimizer satisfies $\rho(\nabla F^{base}(\boldsymbol{\omega}^\star)) < 1$, then for $\boldsymbol{\omega}_0$ in a neighborhood of $\boldsymbol{\omega}^\star$, the iterates $\boldsymbol{\omega}_t$ of Lookahead–Minmax converge to $\boldsymbol{\omega}^\star$ as $t \rightarrow \infty$.*

**Re-visiting the above example.** For the GDA operator we have $\rho(\nabla F^{GDA}(\boldsymbol{\omega}^\star)) = |\lambda| = 1 + \eta^2 \geq 1$ and as $\lambda \triangleq re^{i\theta} \in \mathbb{C}$, with $r > 1$. Applying it $k$ times will thus result in eigenvalues $\{\lambda^{(t)}\}_{t=1}^k = re^{i\theta}, \ldots, r^k e^{ik\theta}$ which are *diverging* and *rotating* in $\mathbb{C}$. The spectral radius of LA–GDA with $k = 2$ is then $1 - \alpha + \alpha\lambda^2$ (see § B) where $\alpha$ selects a particular point that lies between $1 + 0i$ and $\lambda^2$ in $\mathbb{C}$, allowing for reducing the spectral radius of the GDA base operator.

### 4.2 MOTIVATING EXAMPLE: THE BILINEAR GAME

We argue that Lookahead-minmax allows for improved stability and performance on minmax problems due to two main reasons: (i) it handles well rotations, as well as (ii) it reduces the noise due to making more conservative steps. In the following, we disentangle the two, and show in § 4.2.1 that Lookahead-minmax converges fast in the full-batch setting, without presence of noise as each parameter update uses the full dataset.

In § 4.2.2 we consider the challenging problem of Chavdarova et al. (2019), designed to have high variance. We show that besides therein proposed *Stochastic Variance Reduced Extragradient* (SVRE), Lookahead-minmax is the only method that converges on this experiment, while considering all methods of Gidel et al. (2019a, §7.1). More precisely, we consider the following bilinear problem:

$$\min_{\boldsymbol{\theta}\in\mathbb{R}^d}\max_{\boldsymbol{\varphi}\in\mathbb{R}^d}\mathcal{L}(\boldsymbol{\theta},\boldsymbol{\varphi}) = \min_{\boldsymbol{\theta}\in\mathbb{R}^d}\max_{\boldsymbol{\varphi}\in\mathbb{R}^d}\frac{1}{n}\sum_{i=1}^{n}(\boldsymbol{\theta}^\top\boldsymbol{b}_i + \boldsymbol{\theta}^\top\boldsymbol{A}_i\boldsymbol{\varphi} + \boldsymbol{c}_i^\top\boldsymbol{\varphi}), \quad\text{(SB–G)}$$

with $\boldsymbol{\theta},\boldsymbol{\varphi},\boldsymbol{b}_i,\boldsymbol{c}_i\in\mathbb{R}^d$ and $\boldsymbol{A}\in\mathbb{R}^{n\times d\times d}$, $n{=}d{=}100$, and we draw $[\boldsymbol{A}_i]_{kl}{=}\delta_{kli}$ and $[\boldsymbol{b}_i]_k, [\boldsymbol{c}_i]_k{\sim}$ $\mathcal{N}(0, 1/d)$, $1 \le k,l \le d$, where $\delta_{kli}{=}1$ if $k{=}l{=}i$, and 0 otherwise. As one pass we count a forward and backward pass § D.5, and all results are normalized. See § D.1 for hyperparameters.

#### 4.2.1 THE FULL-BATCH SETTING

In Fig. 3 we compare: (i) *GDA* with learning rate $\eta = 10^{-4}$ and $\eta = 0.3$ (in blue), which oscilates around the optimum with small enough step size, and diverges otherwise; (ii) *Unroll-Y* where the max-player is unrolled $k$ steps, before updating the min player, as in (Metz et al., 2017); (iii) *Unroll-XY* where both the players are unrolled $k$ steps with fixed opponent, and the actual updates are done with un unrolled opponent (see § D); (iv) *LA–GDA* with $\alpha = 0.5$ and $\alpha = 0.4$ (in red and pink, resp.) which combines Alg. 1 with GDA. (v) *ExtraGradient*–Eq. EG; (vi) *LA–ExtraGrad*, which combines Alg. 1 with ExtraGradient; (vii) *OGDA* Optimistic GDA (Rakhlin & Sridharan, 2013), as well as (viii) *LA–OGDA* which combines Alg. 1 with OGDA. See § D for description of the algorithms and their implementation. Interestingly, we observe that Lookahead–Minmax allows GDA to converge on this example, and moreover speeds up the convergence of ExtraGradient.

#### 4.2.2 THE STOCHASTIC SETTING

In this section, we show that besides SVRE, Lookahead–minmax also converges on equation SB–G. In addition, we test all the methods of Gidel et al. (2019a, §7.1) using minibatches of several sizes $B = 1, 16, 64$, and sampling without replacement. In particular, we tested: (i) the *Adam* method combined with GDA (shown in blue); (ii) *ExtraGradient*–Eq. EG; as well as (iii) *ExtraAdam* proposed by (Gidel et al., 2019a); (iv) our proposed method *LA-GDA* (Alg. 1) combined with GDA; as well as (v) *SVRE* (Chavdarova et al., 2019, Alg.1) for completeness. Fig. 4 depicts our results. See § D for details on the implementation and choice of hyperparameters. We observe that besides the good performance of LA-GDA on games in the batch setting, it also has the property to cope well large variance of the gradient estimates, and it converges *without* using restarting.

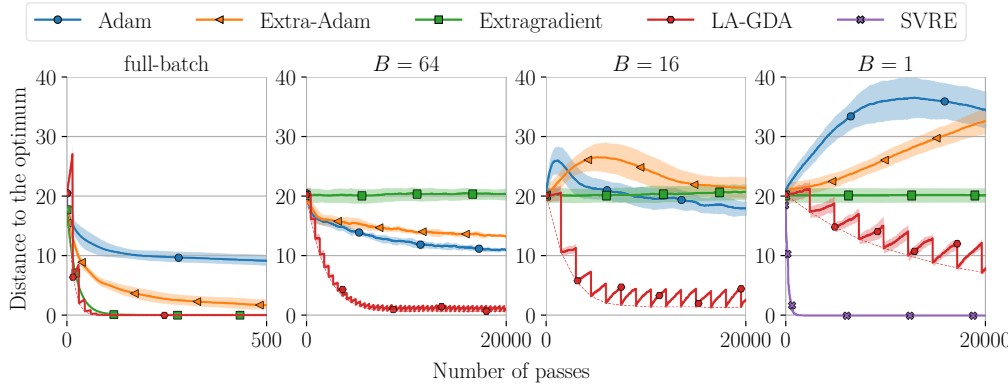

Figure 4: Convergence of *Adam*, *ExtraAdam*, *Extragradient*, *SVRE* and *LA-GDA*, on equation SB–G, for several minibatch sizes $B$, averaged over 5 runs–with random initialization of both the parameters and the data points ($\boldsymbol{A}$, $\boldsymbol{b}$ and $\boldsymbol{c}$). Fast ($\tilde{\boldsymbol{\varphi}},\tilde{\boldsymbol{\theta}}$) and slow ($\boldsymbol{\varphi},\boldsymbol{\theta}$) weights of LA–GDA are shown with solid and dashed lines, resp.

| (32×32) ImageNet | Fréchet Inception distance ↓ | | | Inception score ↑ | | |
|---|---|---|---|---|---|---|
| | no avg | EMA | EMA-slow | no avg | EMA | EMA-slow |
| AltGAN | $15.63 \pm .46$ | $14.16 \pm .27$ | – | $7.23 \pm .13$ | $7.81 \pm .07$ | – |
| LA–AltGAN | $14.37 \pm .20$ | $\mathbf{13.06 \pm .20}$ | $\mathbf{12.53 \pm .06}$ | $7.58 \pm .07$ | $\mathbf{7.97 \pm .11}$ | $\mathbf{8.42 \pm .11}$ |
| NLA–AltGAN | $\mathbf{13.14 \pm .25}$ | $\mathbf{13.07 \pm .25}$ | $12.71 \pm .13$ | $\mathbf{7.85 \pm .05}$ | $7.87 \pm .01$ | $8.10 \pm .05$ |
| ExtraGrad | $15.48 \pm .44$ | $14.15 \pm .63$ | – | $7.31 \pm .06$ | $7.85 \pm .10$ | – |
| LA–ExtraGrad | $14.53 \pm .27$ | $14.13 \pm .23$ | $14.09 \pm .28$ | $7.62 \pm .06$ | $7.70 \pm .07$ | $7.89 \pm .04$ |
| NLA–ExtraGrad | $15.05 \pm .96$ | $14.79 \pm .93$ | $13.88 \pm .45$ | $7.39 \pm .12$ | $7.48 \pm .12$ | $7.76 \pm .15$ |
| **CIFAR-10** | | | | | | |
| AltGAN | $21.37 \pm 1.60$ | $16.92 \pm 1.16$ | – | $7.41 \pm .16$ | $8.03 \pm .13$ | – |
| LA–AltGAN | $16.74 \pm .46$ | $\mathbf{13.98 \pm .47}$ | $\mathbf{12.67 \pm .57}$ | $\mathbf{8.05 \pm .43}$ | $\mathbf{8.19 \pm .05}$ | $\mathbf{8.55 \pm .04}$ |
| ExtraGrad | $18.49 \pm .99$ | $15.47 \pm 1.82$ | – | $7.61 \pm .07$ | $8.05 \pm .09$ | – |
| LA–ExtraGrad | $\mathbf{15.25 \pm .30}$ | $14.68 \pm .30$ | $13.39 \pm .23$ | $\mathbf{7.99 \pm .03}$ | $8.04 \pm .04$ | $8.40 \pm .05$ |
| Unrolled–GAN | $21.04 \pm 1.08$ | $17.51 \pm 1.08$ | – | $7.43 \pm .07$ | $7.88 \pm .12$ | – |
| **SVHN** | | | | | | |
| AltGAN | $7.84 \pm 1.21$ | $6.83 \pm 2.88$ | – | $3.10 \pm .09$ | $\mathbf{3.19 \pm .09}$ | – |
| LA–AltGAN | $3.87 \pm .09$ | $\mathbf{3.28 \pm .09}$ | $3.21 \pm .19$ | $3.16 \pm .02$ | $\mathbf{3.22 \pm .08}$ | $\mathbf{3.30 \pm .07}$ |
| ExtraGrad | $4.08 \pm .11$ | $\mathbf{3.22 \pm .09}$ | – | $\mathbf{3.21 \pm .02}$ | $3.16 \pm .02$ | – |
| LA–ExtraGrad | $\mathbf{3.20 \pm .09}$ | $\mathbf{3.16 \pm .14}$ | $\mathbf{3.15 \pm .31}$ | $\mathbf{3.20 \pm .02}$ | $3.19 \pm .03$ | $3.20 \pm .04$ |

Table 1: Comparison of LA–GAN with its respective baselines AltGAN and ExtraGrad (see § 5.1 for naming), using FID (lower is better) and IS (higher is better), and *best obtained scores*. EMA denotes *exponential moving average*, see § F. All methods use *Adam*, see § G for detailed description. Results are averaged over 5 runs. We run each experiment for 500K iterations. See § H and § 5.2 for details on architectures and hyperparameters and for discussion on the results, resp. Our overall best obtained FID scores are 12.19 on CIFAR-10 and 2.823 on SVHN, see § I for samples of these generators. Best scores obtained for each metric and dataset are highlighted in yellow. For each column the best score is in bold along with any score within its standard deviation reach.

| | Unconditional GANs | | | | | | Conditional GANs | |
|---|---|---|---|---|---|---|---|---|
| | SNGAN | Prog.GAN | NCSN | WS-SVRE | ExtraAdam | LA-AltGAN | SNGAN | BigGAN |
| | Miyato et al. | Karras et al. | Song & Ermon | Chavdarova et al. | Gidel et al. | (ours) | Miyato et al. | Brock et al. |
| FID | 21.7 | – | 25.32 | 16.77 | $16.78 \pm .21$ | **12.19** | 25.5 | 14.73 |
| IS | 8.22 | $8.80 \pm .05$ | 8.87 | – | $8.47 \pm .10$ | 8.78 | 8.60 | **9.22** |

Table 2: Summary of the recently reported best scores on CIFAR-10 and benchmark with LA–GAN, *using published results*. Note that the architectures are *not* identical for all the methods–see § 5.2.

# 5 EXPERIMENTS

In this section, we empirically benchmark Lookahead–minmax–Alg. 1 for *training GANs*. For the purpose of fair comparison, as an *iteration* we count each update of *both* the players, see Alg. 3.

## 5.1 EXPERIMENTAL SETUP

**Datasets.** We used the following image datasets: (i) **MNIST** (Lecun & Cortes, 1998), (ii) **CI-FAR-10** (Krizhevsky, 2009, §3), (iii) **SVHN** (Netzer et al., 2011), and (iv) **ImageNet** ILSVRC 2012 (Russakovsky et al., 2015), using resolution of $28 \times 28$ for **MNIST**, and $3 \times 32 \times 32$ for the rest.

**Metrics.** We use **Inception score** (IS, Salimans et al., 2016) and **Fréchet Inception distance** (FID, Heusel et al., 2017) as these are most commonly used metrics for image synthesis, see § H.1 for details. On datasets other than ImageNet, IS is less consistent with the sample quality (see H.1.1).

**DNN architectures.** For experiments on **MNIST**, we used the DCGAN architectures (Radford et al., 2016), described in § H.2.1. For SVHN and CIFAR-10, we used the ResNet architectures, replicating the setup in (Miyato et al., 2018; Chavdarova et al., 2019), described in detail in H.2.2.

**Optimization methods.** We use: (i) **AltGan:** the standard alternating GAN, (ii) **ExtraGrad:** the extragradient method, as well as (iii) **UnrolledGAN:** (Metz et al., 2017). We combine Lookahead-minmax with (i) and (ii), and we refer to these as **LA–AltGAN** and **LA–ExtraGrad**, respectively or for both as **LA–GAN** for brevity. We denote *nested* LA–GAN with prefix **NLA**, see § G.1. All methods in this section use *Adam* (Kingma & Ba, 2015). We compute Exponential Moving Average (EMA, see def. in § F) of both the fast and slow weights–called **EMA** and **EMA–slow**, resp. See Appendix for results of uniform iterate averaging and *RAdam* (Liu et al., 2020).

## 5.2 RESULTS AND DISCUSSION

**Comparison with baselines.** Table 1 summarizes our comparison of combining Alg. 1 with AltGAN and ExtraGrad. On all datasets, we observe that the iterates (column "no avg") of LA–AltGAN and LA–ExtraGrad perform notably better than the corresponding baselines, and using EMA on LA–AltGAN and LA–ExtraGrad further improves the FID and IS scores obtained with LA–AltGAN. As the performances improve after each LA–step, see Fig. 2 computing EMA solely on the slow weights further improves the scores. It is interesting to note that on most of the datasets, the scores of the iterates of LA–GAN (column *no-avg*) outperform the EMA scores of the respective baselines. In some cases, EMA for AltGAN does not provide improvements, as the iterates diverge relatively early. In our baseline experiments, ExtraGrad outperforms AtlGAN–while requiring twice the computational time of the latter per iteration. The addition of Lookahead–minimax stabilizes AtlGAN making it competitive to LA–ExtraGrad while using *half* of the computational time.

In Table 3 we report our results on MNIST, where–different from the rest of the datasets–the training of the baselines is stable, to gain insights if LA–GAN still yields improved performances. The best FID scores of the iterates (column "no avg") are obtained with LA–GAN. Interestingly, although we obtain that the last iterates are not LSSP (which could be due to stochasticity), from Fig. 5–which depicts the eigenvalues of JVF, we observe that after convergence LA–GAN shows no rotations.

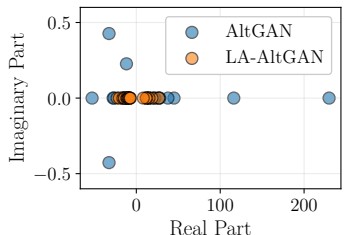

Figure 5: Eigenvalues of $v'(\boldsymbol{\theta}, \boldsymbol{\varphi})$ at 100K iterations on MNIST, see § 5.2.

|  | no avg | EMA | EMA-slow |
|---|---|---|---|
| AltGAN | $.094 \pm .006$ | $\mathbf{.031 \pm .002}$ | – |
| LA–AltGAN | $\mathbf{.053 \pm .004}$ | $.029 \pm .002$ | $.032 \pm .002$ |
| ExtraGrad | $.094 \pm .013$ | $.032 \pm .003$ | – |
| LA–ExtraGrad | $\mathbf{.053 \pm .005}$ | $.032 \pm .002$ | $.034 \pm .001$ |
| Unrolled–GAN | $.077 \pm .006$ | $\mathbf{.030 \pm .002}$ | – |

Table 3: FID (lower is better) results on MNIST, averaged over 5 runs. Each experiment is trained for 100K iterations. Note that Unrolled–GAN is computationally more expensive: in the order of the ratio $4 : 22$–as we used 20 steps of unrolling what gave best results. See 5.2 & H for implementation and discussion, resp.

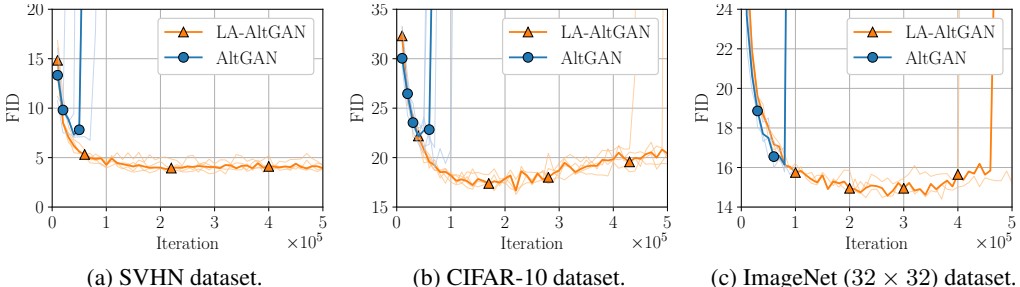

(a) SVHN dataset.

(b) CIFAR-10 dataset.

(c) ImageNet ($32 \times 32$) dataset.

Figure 6: Improved stability of LA–AltGAN relative to its baselines on SVHN, CIFAR-10 and ImageNet, over 5 runs. The median and the individual runs are illustrated with ticker solid lines and with transparent lines, respectively. See § 5.2 for discussion.

**Benchmark on CIFAR-10 using reported results.** Table 2 summarizes the recently reported *best* obtained FID and IS scores on CIFAR-10. Although using the class labels–Conditional GAN is known to improve GAN performances (Radford et al., 2016), we outperform BigGAN (Brock et al., 2019) on CIFAR-10. Notably, our model and BigGAN have $5.1$M and $158.3$M parameters in total, respectively, and we use minibatch size of $128$, whereas BigGAN uses $2048$ samples. The competitive result of (Yazıcı et al., 2019) of average $12.56$ FID uses $\sim 3.5$ times larger model than ours and is omitted from Table 2 as it does not use the standard metrics pipeline.

**Additional memory & computational cost.** Lookahead-minmax requires the same extra memory footprint as EMA and uniform averaging (one additional set of parameters per player)—both of which are updated each step whereas LA–GAN is updated *once every $k$ iterations*.

**On the choice of $\alpha$ and $k$.** We fixed $\alpha = 0.5$, and we experimented with several values of $k$–while once selected, keeping $k$ fixed throughout the training. We observe that all values of $k$ improve the baseline. Using larger $k$ in large part depends on the stability of the base-optimizer: if it quickly diverges, smaller $k$ is necessary to stabilize the training. Thus we used $k = 5$ and $k = 5000$ for LA–

AltGAN and LA–ExtraGrad, resp. When using larger values of $k$ we noticed that the obtained scores would periodically drastically improve every $k$ iterations–after an LA step, see Fig.2. Interestingly, complementary to the results of (Berard et al., 2020)–who locally analyze the vector field, Fig.2 confirms the presence of rotation *while taking into account the used optimizer*, and empirically validates the geometric justification of Lookahead–minmax illustrated in Fig.1. The necessity of small k for unstable baselines motivates *nested* LA–Minmax using two different values of $k$, of which one is relatively small (the inner LA–step) and the other is larger (the outer LA–step). Intuitively, the inner and outer LA–steps in such nested LA–GAN tackle the variance and the rotations, resp.

**Stability of convergence.** Fig. 6 shows that LA–GAN *consistently improved the stability of its respective baseline*. Despite using $1 : 5$ update ratio for $G : D$–known to improve stability, our baselines always diverge (also reported by Chavdarova et al., 2019; Brock et al., 2019). On the other hand, LA–GAN diverged only few times and notably later in the training relative to the same baseline, see additional results in § I. The stability further improves using nested LA–steps as in Fig.2.

# 6 RELATED WORK

**Parameter averaging.** In the context of convex single-objective optimization, taking an arithmetic average of the parameters as by Polyak & Juditsky (1992); Ruppert (1988) is well-known to yield faster convergence for convex functions and allowing the use of larger constant step-sizes in the case of stochastic optimization (Dieuleveut et al., 2017). It recently gained more interest in deep learning in general (Garipov et al., 2018), in natural language processing (Merity et al., 2018), and particularly in GANs (Yazıcı et al., 2019) where researchers report the performance of a uniform or exponential moving average of the iterates. Such averaging as a post-processing after training is fundamentally different from immediately applying averages during training. Lookahead as of our interest here in spirit is closer to extrapolation methods (Korpelevich, 1976) which rely on gradients taken not at the current iterate but at an extrapolated point for the current trajectory. For highly complex optimization landscapes such as in deep learning, the effect of using gradients at perturbations of the current iterate has a desirable smoothing effect which is known to help training speed and stability in the case of non-convex single-objective optimization (Wen et al., 2018; Haruki et al., 2019).

**GANs.** Several proposed methods for GANs are motivated by the "recurrent dynamics". Apart from the already introduced works, (i) Yadav et al. (2018) use prediction steps, (ii) Daskalakis et al. (2018) propose *Optimistic Mirror Decent* (OMD) (iii) Balduzzi et al. (2018) propose the *Symplectic Gradient Adjustment* (SGA) (iv) Gidel et al. (2019b) propose negative momentum, (v) Xu et al. (2020) propose closed-loop based method from control theory, among others. Besides its simplicity, the key benefit of LA–GAN is that it handles well *both* the rotations of the vector field as well as noise from stochasticity, thus performing well on real-world applications.

# 7 CONCLUSION

Motivated by the adversarial component of games and the negative impact of noise on games, we proposed an extension of the Lookahead algorithm to games, called "Lookahead–minmax". On the bilinear toy example we observe that combining Lookahead–minmax with standard gradient methods converges, and that Lookahead–minmax copes very well with the high variance of the gradient estimates. Exponential moving averaging of the iterates is known to help obtain improved performances for GANs, yet it does not impact the actual iterates, hence does not stop the algorithm from (early) divergence. Lookahead–minmax goes beyond such averaging, requires less computation than running averages, and it is *straightforward to implement*. It can be applied to any optimization method, and in practice it *consistently* improves the stability of its respective baseline. While we do *not* aim at obtaining a new state-of-the-art result, it is remarkable that Lookahead–minmax obtains competitive result on CIFAR–10 of 12.19 FID–outperforming the 30–times larger BigGAN.

As Lookahead–minmax uses two additional hyperparameters, future directions include developing adaptive schemes of obtaining these coefficients throughout training, which could further speed up the convergence of Lookahead–minmax.

ACKNOWLEDGMENTS

This work was in part done while TC and FF were affiliated with Idiap research institute. TC was funded in part by the grant 200021-169112 from the Swiss National Science Foundation, and MP was funded by the grant 40516.1 from the Swiss Innovation Agency. TC would like to thank Hugo Berard for insightful discussions on the optimization landscape of GANs and sharing their code, as well as David Balduzzi for insightful discussions on $n$-player differentiable games.

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

## A  THEORETICAL ANALYSIS OF LOOKAHEAD FOR SINGLE OBJECTIVE MINIMIZATION—PROOF OF THEOREM 1

In this section we give the theoretical justification for the convergence results claimed in Section 3, in Theorem 1 and Remark 1. We consider the LA optimizer, with $k$ SGD prediction steps, that is,

$$\tilde{\boldsymbol{\omega}}_{t+k} = \tilde{\boldsymbol{\omega}}_t - \gamma \sum_{i=1}^{k} g(\tilde{\boldsymbol{\omega}}_{t+i-1})$$

where $g\colon \mathcal{X} \to \mathbb{R}$ is a stochastic gradient estimator, $\mathbb{E}g(\boldsymbol{\omega}) = \nabla f(\boldsymbol{\omega})$, with bounded variance $\mathbb{E}\|g(\boldsymbol{\omega})\|^2 \leq \sigma^2$. Further, we assume that $f$ is $L$-smooth for a constant $L \geq 0$ (but not necessarily convex). We count the total number of gradient evaluations $K$. As is standard, we consider as output of the algorithm a uniformly at random chosen iterate $\boldsymbol{\omega}_{\text{out}}$ (we prove convergence of $\mathbb{E}\|\nabla f(\boldsymbol{\omega})\|^2 = \|\nabla f(\boldsymbol{\omega}_{\text{out}})\|^2$. This matches the setting considered in (Wang et al., 2020).

Wang et al. (2020, Theorem 2) show that if the stepsize $\gamma$ satisfies

$$\alpha\gamma L + (1-\alpha)^2\gamma^2 L^2 k(k-1) \leq 1, \tag{1}$$

then the expected squared gradient norm of the LA iterates after $T$ updates of the slow sequence, i.e. $K = kT$ gradient evaluations, can be bounded as

$$\mathcal{O}\left(\frac{F_0}{\alpha\gamma K} + \alpha\gamma L\sigma^2 + (1-\alpha)^2\gamma^2 L^2\sigma^2(k-1)\right) \tag{2}$$

where $F_0 := f(\omega_0) - f_{\text{inf}}$ denotes an upper bound on the on the optimality gap. Now, departing from Wang et al. (2020), we can directly derive the claimed bound in Theorem 1 by choosing $\gamma$ such as to minimze equation 2, while respecting the constraints given in equation 1 (two necessary constraints are e.g. $\gamma^2 \leq \frac{1}{(1-\alpha)^2 L^2 k(k-1)}$ and $\gamma \leq \frac{1}{\alpha L}$). For this minimization with respect to $\gamma$, see for instance (Koloskova et al., 2020, Lemma 15), where we plug in $r_0 = \frac{F_0}{\alpha}$, $b = \alpha L\sigma^2$, $e = (1-\alpha)^2 L^2\sigma^2(k-1)$ and $d = \min\{\frac{1}{\alpha L}, \frac{1}{(1-\alpha)Lk}\}$.

The improved result for the quadratic case follows from the observations in Woodworth et al. (2020b) who analyze local SGD on quadratic functions (they show that linear update algorithms (such as local SGD on quadratics) benefit from variance reduction, allowing to recover the same convergence guarantees as single-threaded SGD. These observations also hold for non-convex quadratic functions.

We point out, that the the choice of the stepsize $\gamma = \frac{1}{\sqrt{kT}}$ proposed in Theorem 2 in (Wang et al., 2020) does not necessarily satisfy their constraint given in equation 1 for small values of $T$. Whilst their conclusions remain valid when $k$ is a constant (or in the limit, $T \to \infty$), the current analyses (including our slightly improved bound) do not show that LA can in general match the performance upper bounds of SGD when $k$ is large. Whilst casting LA as an instance of local SGD was useful to derive the first general convergence guarantees, we believe—in regard to recently derived lower bounds (Woodworth et al., 2020a) that show limitations on convex functions—that this approach is limited and more carefully designed analyses are required to derive tighter results for LA in future works.

## B  PROOF OF THEOREM 3

*Proof.* Let $F^{base}(\boldsymbol{\omega})$ denote the operator that an optimization method applies to the iterates $\boldsymbol{\omega} = (\boldsymbol{\theta}, \boldsymbol{\varphi})$. Let $\boldsymbol{\omega}^\star$ be a stable fixed point of $F^{base}$ i.e. $\rho(\nabla F^{base}(\boldsymbol{\omega}^\star)) < 1$. Then, the operator of Lookahead-Minmax algorithhm $F^{LA}$ (when combined with $F^{base}(\cdot)$) is:

$$F^{LA}(\boldsymbol{\omega}) = \boldsymbol{\omega} + \alpha((F^{base})^k(\boldsymbol{\omega}) - \boldsymbol{\omega})$$
$$= (1-\alpha)\boldsymbol{\omega} + \alpha(F^{base})^k(\boldsymbol{\omega}),$$

with $\alpha \in [0, 1]$ and $k \in \mathbb{N}, k \geq 2$.

Let $\lambda$ denote the eigenvalue of $J^{base} \triangleq \nabla F^{base}(\boldsymbol{\omega}^\star)$ with largest modulus, *i.e.* $\rho(\nabla F^{base}(\boldsymbol{\omega}^\star)) = |\lambda|$, let $\boldsymbol{u}$ be its associated eigenvector: $J^{base}\boldsymbol{u} = \lambda\boldsymbol{u}$.

Using the fixed point property ($F(\boldsymbol{\omega}^\star) = \boldsymbol{\omega}^\star$), the Jacobian of Lookahead-Minmax at $\boldsymbol{\omega}^\star$ is thus:

$$J^{LA} = \nabla F^{LA}(\boldsymbol{\omega}^\star) = (1-\alpha)I + \alpha(J^{base})^k \,.$$

By noticing that:

$$
\begin{aligned}
J^{LA}\boldsymbol{u} &= ((1-\alpha)I + \alpha(J^{base})^k)\boldsymbol{u} \\
&= ((1-\alpha) + \alpha\lambda^k)\boldsymbol{u} \,,
\end{aligned}
$$

we deduce $\boldsymbol{u}$ is an eigenvector of $J^{LA}$ with eigenvalue $1 - \alpha + \alpha\lambda^k$.

We know that $\rho(J^{base}) = \max\{|\lambda_0^{base}|, ..., |\lambda_n^{base}|\} < 1$ (by the assumption of this theorem). To prove Theorem 3 by contradiction, let us assume that: $\rho(J^{LA}) \geq 1$, what implies that there exist an eigenvalue of the spectrum of $J^{LA}$ such that:

$$\exists \lambda^{LA} \in Spec(J^{LA}) \quad s.t. \quad |\lambda^{LA}| \triangleq |1 - \alpha + \alpha(\lambda^{base})^k| \geq 1 \,. \tag{3}$$

As $|\lambda^{base}| < 1$, we have $|\lambda^{base}|^k = |(\lambda^{base})^k| < 1$. Furthermore, the set of points $\{p \in \mathbb{C} | \alpha \in [0,1], p = 1 - \alpha + \alpha(\lambda^{base})^k\}$ is describing a segment in the complex plane between $(\lambda^{base})^k$ and $1 + 0i$ (excluded from the segment). As both ends of the segment are in the unit circle, it follows that the left hand side of the inequality of equation 3 is strictly smaller than 1. By contradiction, this implies $\rho(J^{LA}) < 1$, hence it converges linearly. $\square$

Hence, if $\boldsymbol{\omega}^\star$ is a stable fixed point for some inner optimizer whose operator satisfies our assumption, then $\boldsymbol{\omega}^\star$ is a stable fixed point for Lookahead-Minmax as well.

## C  SOLUTIONS OF A MIN-MAX GAME

In § 2 we briefly mention the "ideal" convergence point of a 2-player games and defined LSSP–required to follow up our result in Fig. 5. For completeness of this manuscript, in this section we define Nash equilibria formally, and we list relevant works along this line which further question this notion of optimality in games.

For simplicity, in this section we focus on the case when the two players aim at minimizing and maximizing the same value function $\mathcal{L}^\varphi = -\mathcal{L}^\theta := \mathcal{L}$, called *zero-sum* or *minimax* game. More precisely:

$$\min_{\boldsymbol{\theta}\in\Theta} \max_{\boldsymbol{\varphi}\in\Phi} \mathcal{L}(\boldsymbol{\theta}, \boldsymbol{\varphi}) \,, \tag{ZS-G}$$

where $\mathcal{L}: \Theta \times \Phi \mapsto \mathbb{R}$.

**Nash Equilibria.**   For 2-player games, ideally we would like to converge to a point called *Nash equilibrium* (NE). In the context of game theory, NE is a combination of strategies from which, no player has an incentive to deviate unilaterally. More formally, Nash equilibria for continuous games is defined as a point $(\boldsymbol{\varphi}^\star, \boldsymbol{\theta}^\star)$ where:

$$\mathcal{L}(\boldsymbol{\varphi}^\star, \boldsymbol{\theta}) \leq \mathcal{L}(\boldsymbol{\varphi}^\star, \boldsymbol{\theta}^\star) \leq \mathcal{L}(\boldsymbol{\varphi}, \boldsymbol{\theta}^\star) \,. \tag{NE}$$

Such points are (locally) optimal for both players with respect to their own decision variable, *i.e.* no player has the incentive to unilaterally deviate from it.

**Differential Nash Equilibria.**   In machine learning we are interested in differential games where $\mathcal{L}$ is twice differentiable, in which case such NE needs to satisfy slightly stronger conditions. A point $(\boldsymbol{\theta}^\star, \boldsymbol{\varphi}^\star)$ is a *Differential Nash Equilibrium* (DNE) of a zero-sum game iff:

$$
\begin{aligned}
&||\nabla_{\boldsymbol{\theta}}\mathcal{L}(\boldsymbol{\theta}^\star, \boldsymbol{\varphi}^\star)|| = ||\nabla_{\boldsymbol{\varphi}}\mathcal{L}(\boldsymbol{\theta}^\star, \boldsymbol{\varphi}^\star)|| = 0, \\
&\nabla_{\boldsymbol{\theta}}^2 \mathcal{L}(\boldsymbol{\theta}^\star, \boldsymbol{\varphi}^\star) \succ 0, \text{ and} \\
&\nabla_{\boldsymbol{\varphi}}^2 \mathcal{L}(\boldsymbol{\theta}^\star, \boldsymbol{\varphi}^\star) \prec 0 \,,
\end{aligned}
\tag{DNE}
$$

where $A \succ 0$ and $A \prec 0$ iff $A$ is positive definite and negative definite, respectively. Note that the key difference between DNE and NE is that $\nabla_{\boldsymbol{\theta}}^2 \mathcal{L}(\cdot)$ and $\nabla_{\boldsymbol{\varphi}}^2 \mathcal{L}(\cdot)$ for DNE are required to be definite (instead of semidefinite).

**(Differential) Stackelberg Equilibria.** A recent line of works questions the choice of DNEs as a game solution of multiple machine learning applications, including GANs (Tanner et al., 2020; Jin et al., 2020; Wang et al., 2020). Namely, as GANs are optimized sequentially, authors argue that a so called Stackelberg game is more suitable, which consist of a *leader* player and a *follower*. Namely, the leader can choose her action while knowing the other player plays a best-response:

$$
\begin{aligned}
\text{Leader:} \quad & \boldsymbol{\theta}^\star = \arg\min_{\boldsymbol{\theta} \in \Theta} \left\{ \mathcal{L}(\boldsymbol{\theta}, \boldsymbol{\varphi}) : \boldsymbol{\varphi} = \arg\max_{\boldsymbol{\varphi} \in \Phi} \mathcal{L}(\boldsymbol{\theta}, \boldsymbol{\varphi}) \right\} \\
\text{Follower:} \quad & \boldsymbol{\varphi}^\star = \arg\max_{\boldsymbol{\varphi} \in \Phi} \mathcal{L}(\boldsymbol{\theta}^\star, \boldsymbol{\varphi})
\end{aligned}
\tag{Stackelberg–ZS}
$$

Briefly, such games converge to a so called *Differential Stackelberg Equilibria (DSE)*, defined as a point $(\boldsymbol{\theta}^\star, \boldsymbol{\varphi}^\star)$ where:

$$
\nabla_{\boldsymbol{\theta}} \mathcal{L}(\boldsymbol{\theta}^\star, \boldsymbol{\varphi}^\star) = 0 \Leftrightarrow v(\boldsymbol{\theta}^\star, \boldsymbol{\varphi}^\star) = 0, \text{ and}
\tag{4}
$$

$$
[\nabla_{\boldsymbol{\theta}}^2 \mathcal{L} - \nabla_{\boldsymbol{\theta}} \nabla_{\boldsymbol{\varphi}} \mathcal{L} (\nabla_{\boldsymbol{\varphi}}^2 \mathcal{L})^{-1} (\nabla_{\boldsymbol{\theta}} \nabla_{\boldsymbol{\varphi}} \mathcal{L})^\top](\boldsymbol{\theta}^\star, \boldsymbol{\varphi}^\star) > 0.
\tag{DSE}
$$

**Remark.** (Berard et al., 2020) show that GANs do not converge to DNEs, however, these convergence points often have good performances for the generator. Recent works on DSEs give a game-theoretic interpretation of these solution concepts–which constitute a superset of DNEs. Moreover, while DNEs are not guaranteed to exist, approximate NE are guaranteed to exist (Daskalakis et al., 2020). A more detailed discussion is out of the scope of this paper, and we encourage the interested reader to follow up the given references.

# D   EXPERIMENTS ON THE BILINEAR EXAMPLE

In this section we list the details regarding our implementation of the experiments on the bilinear example of equation SB–G that were presented in § 4.2. In particular: (i) in § D.1 we list the implementational details of the benchmarked algorithms,   (ii) in § D.2 and D.4 we list the hyperparameters used in § 4.2.1 and § 4.2.2, respectively   (iii) in § D.5 we describe the choice of the x-axis of the toy experiments presented in the paper, and finally   (iv) in § D.6 we present visualizations in aim to improve the reader's intuition on how Lookahead-minmax works on games.

## D.1   IMPLEMENTATION DETAILS

**Gradient Descent Ascent (GDA).** We use an alternating implementation of GDA where the players are updated *sequentially*, as follows:

$$
\boldsymbol{\varphi}_{t+1} = \boldsymbol{\varphi}_t - \eta \nabla_{\boldsymbol{\varphi}} \mathcal{L}(\boldsymbol{\theta}_t, \boldsymbol{\varphi}_t), \qquad \boldsymbol{\theta}_{t+1} = \boldsymbol{\theta}_t + \eta \nabla_{\boldsymbol{\theta}} \mathcal{L}(\boldsymbol{\theta}_t, \boldsymbol{\varphi}_{t+1})
\tag{GDA}
$$

**ExtraGrad.** Our implementation of extragradient follows equation EG, with $\mathcal{L}^{\boldsymbol{\theta}}(\cdot) = -\mathcal{L}^{\boldsymbol{\varphi}}(\cdot)$, thus:

$$
\text{Extrapolation:} \begin{cases} \boldsymbol{\theta}_{t+\frac{1}{2}} = \boldsymbol{\theta}_t - \eta \nabla_{\boldsymbol{\theta}} \mathcal{L}(\boldsymbol{\theta}_t, \boldsymbol{\varphi}_t) \\ \boldsymbol{\varphi}_{t+\frac{1}{2}} = \boldsymbol{\varphi}_t + \eta \nabla_{\boldsymbol{\varphi}} \mathcal{L}(\boldsymbol{\theta}_t, \boldsymbol{\varphi}_t) \end{cases} \text{Update:} \begin{cases} \boldsymbol{\theta}_{t+1} = \boldsymbol{\theta}_t - \eta \nabla_{\boldsymbol{\theta}} \mathcal{L}(\boldsymbol{\theta}_{t+\frac{1}{2}}, \boldsymbol{\varphi}_{t+\frac{1}{2}}) \\ \boldsymbol{\varphi}_{t+1} = \boldsymbol{\varphi}_t + \eta \nabla_{\boldsymbol{\varphi}} \mathcal{L}(\boldsymbol{\theta}_{t+\frac{1}{2}}, \boldsymbol{\varphi}_{t+\frac{1}{2}}) \end{cases}.
\tag{EG–ZS}
$$

**OGDA.** Optimistic Gradient Descent Ascent (Denisovich, 1980; Rakhlin & Sridharan, 2013; Daskalakis et al., 2018) has last iterate convergence guaranties when the objective is linear for both the min and the max player. The update rule is as follows:

$$
\begin{cases} \boldsymbol{\theta}_{t+1} = \boldsymbol{\theta}_t - 2\eta \nabla_{\boldsymbol{\theta}} \mathcal{L}(\boldsymbol{\theta}_t, \boldsymbol{\varphi}_t) + \eta \nabla_{\boldsymbol{\theta}} \mathcal{L}(\boldsymbol{\theta}_{t-1}, \boldsymbol{\varphi}_{t-1}) \\ \boldsymbol{\varphi}_{t+1} = \boldsymbol{\varphi}_t + 2\eta \nabla_{\boldsymbol{\varphi}} \mathcal{L}(\boldsymbol{\theta}_t, \boldsymbol{\varphi}_t) - \eta \nabla_{\boldsymbol{\varphi}} \mathcal{L}(\boldsymbol{\theta}_{t-1}, \boldsymbol{\varphi}_{t-1}) \end{cases}.
\tag{OGDA}
$$

Contrary to EG–ZS, OGDA requires one gradient computation per parameter update, and requires storing the previously computed gradient at $t-1$. Interestingly, both EG–ZS and OGDA can be seen as an approximations of the proximal point method for min-max problem, see (Mokhtari et al., 2019) for details.

**Unroll-Y.** Unrolling was introduced by Metz et al. (2017) as a way to mitigate mode collapse of GANs. It consists of finding an optimal max–player $\varphi^\star$ for a fixed min–player $\boldsymbol{\theta}$, *i.e.* $\varphi^\star(\boldsymbol{\theta}) = \arg\max_{\varphi} \mathcal{L}^{\varphi}(\boldsymbol{\theta}, \varphi)$ through "unrolling" as follows:

$$\varphi_t^0 = \varphi_t, \qquad \varphi_t^{m+1}(\boldsymbol{\theta}) = \varphi_t^m - \eta\nabla_{\varphi}\mathcal{L}^{\varphi}(\boldsymbol{\theta}_t, \varphi_t^m), \qquad \varphi_t^\star(\boldsymbol{\theta}_t) = \lim_{m\to\infty} \varphi_t^m(\boldsymbol{\theta}).$$

In practice $m$ is a finite number of unrolling steps, yielding $\varphi_t^m$. The min–player $\boldsymbol{\theta}_t$, *e.g.* the generator, can be updated using the unrolled $\varphi_t^m$, while the update of $\varphi_t$ is unchanged:

$$\boldsymbol{\theta}_{t+1} = \boldsymbol{\theta}_t - \eta\nabla_{\boldsymbol{\theta}}\mathcal{L}^{\boldsymbol{\theta}}(\boldsymbol{\theta}_t, \varphi_t^m), \qquad \varphi_{t+1} = \varphi_t - \eta\nabla_{\varphi}\mathcal{L}^{\varphi}(\boldsymbol{\theta}_t, \varphi_t) \tag{UR–X}$$

**Unroll-XY.** While Metz et al. (2017) only unroll one player (the discriminator in their GAN setup), we extended the concept of unrolling to games and for completeness also considered unrolling *both* players. For the bilinear experiment we also have that $\mathcal{L}^{\boldsymbol{\theta}}(\boldsymbol{\theta}_t, \varphi_t) = -\mathcal{L}^{\varphi}(\boldsymbol{\theta}_t, \varphi_t)$.

**Adam.** Adam (Kingma & Ba, 2015) computes an exponentially decaying average of both past gradients $m_t$ and squared gradients $v_t$, for each parameter of the model as follows:

$$m_t = \beta_1 m_{t-1} + (1-\beta_1)g_t \tag{5}$$
$$v_t = \beta_2 v_{t-1} + (1-\beta_2)g_t^2, \tag{6}$$

where the hyperparameters $\beta_1, \beta_2 \in [0,1]$, $m_0 = 0$, $v_0 = 0$, and $t$ denotes the iteration $t = 1, \ldots T$. $m_t$ and $v_t$ are respectively the estimates of the first and the second moments of the stochastic gradient. To compensate the bias toward 0 due to their initialization to $m_0 = 0$, $v_0 = 0$, Kingma & Ba (2015) propose to use bias-corrected estimates of these first two moments:

$$\hat{m}_t = \frac{m_t}{1-\beta_1^t} \tag{7}$$
$$\hat{v}_t = \frac{v_t}{1-\beta_2^t}. \tag{8}$$

Finally, the Adam update rule for all parameters at $t$-th iteration $\boldsymbol{\omega}_t$ can be described as:

$$\boldsymbol{\omega}_{t+1} = \boldsymbol{\omega}_t - \eta\frac{\hat{\boldsymbol{m}}_t}{\sqrt{\hat{\boldsymbol{v}}_t} + \varepsilon}. \tag{Adam}$$

**Extra-Adam.** Gidel et al. (2019a) adjust Adam for extragradient equation EG and obtain the empirically motivated *ExtraAdam* which re-uses the same running averages of equation Adam when computing the extrapolated point $\boldsymbol{\omega}_{t+\frac{1}{2}}$ as well as when computing the new iterate $\boldsymbol{\omega}_{t+1}$ (see Alg.4, Gidel et al., 2019a). We used the provided implementation by the authors.

**SVRE.** Chavdarova et al. (2019) propose SVRE as a way to cope with variance in games that may cause divergence otherwise. We used the restarted version of SVRE as used for the problem of equation SB–G described in (Alg3, Chavdarova et al., 2019), which we describe in Alg. 2 for completeness–where $d_{\boldsymbol{\theta}}$ and $d_{\varphi}$ denote "variance corrected" gradient:

$$\boldsymbol{d}_{\varphi}(\boldsymbol{\theta}, \varphi, \boldsymbol{\theta}^{\mathcal{S}}, \varphi^{\mathcal{S}}) := \boldsymbol{\mu}_{\varphi} + \nabla_{\varphi}\mathcal{L}^{\varphi}(\boldsymbol{\theta}, \varphi, \mathcal{D}[n_d], \mathcal{Z}[n_z]) - \nabla_{\varphi}\mathcal{L}^{\varphi}(\boldsymbol{\theta}^{\mathcal{S}}, \varphi^{\mathcal{S}}, \mathcal{D}[n_d], \mathcal{Z}[n_z]) \quad (9)$$
$$\boldsymbol{d}_{\boldsymbol{\theta}}(\boldsymbol{\theta}, \varphi, \boldsymbol{\theta}^{\mathcal{S}}, \varphi^{\mathcal{S}}) := \boldsymbol{\mu}_{\boldsymbol{\theta}} + \nabla_{\boldsymbol{\theta}}\mathcal{L}^{\boldsymbol{\theta}}(\boldsymbol{\theta}, \varphi, \mathcal{Z}[n_z]) - \nabla_{\boldsymbol{\theta}}\mathcal{L}^{\boldsymbol{\theta}}(\boldsymbol{\theta}^{\mathcal{S}}, \varphi^{\mathcal{S}}, \mathcal{Z}[n_z]), \quad (10)$$

where $\boldsymbol{\theta}^{\mathcal{S}}$ and $\varphi^{\mathcal{S}}$ are the snapshots and $\boldsymbol{\mu}_{\boldsymbol{\theta}}$ and $\boldsymbol{\mu}_{\varphi}$ their respective gradients. $\mathcal{D}$ and $\mathcal{Z}$ denote the finite data and noise datasets. With a probability $p$ (fixed) before the computation of $\boldsymbol{\mu}_{\varphi}^{\mathcal{S}}$ and $\boldsymbol{\mu}_{\boldsymbol{\theta}}^{\mathcal{S}}$, we decide whether to restart SVRE (by using the averaged iterate as the new starting point–Alg. 2, Line 6–$\bar{\boldsymbol{\omega}}_t$) or computing the batch snapshot at a point $\boldsymbol{\omega}_t$. For consistency, we used the provided implementation by the authors.

---

**Algorithm 2** Pseudocode for Restarted SVRE.

---

1: **Input:** Stopping time $T$, learning rates $\eta_{\boldsymbol{\theta}}, \eta_{\boldsymbol{\varphi}}$, losses $\mathcal{L}^{\boldsymbol{\theta}}$ and $\mathcal{L}^{\boldsymbol{\varphi}}$, probability of restart $p$, dataset $\mathcal{D}$, noise dataset $\mathcal{Z}$, with $|\mathcal{D}| = |\mathcal{Z}| = n$.
2: **Initialize:** $\boldsymbol{\varphi}, \boldsymbol{\theta}, t = 0$        $\triangleright$ $t$ is for the online average computation.
3: **for** $e = 0$ **to** $T-1$ **do**
4:     Draw `restart` $\sim \mathrm{B}(p)$.        $\triangleright$ Check if we restart the algorithm.
5:     **if** `restart` **and** $e > 0$ **then**
6:        $\boldsymbol{\varphi} \leftarrow \bar{\boldsymbol{\varphi}}, \ \ \boldsymbol{\theta} \leftarrow \bar{\boldsymbol{\theta}}$ and $t = 1$
7:     **end if**
8:     $\boldsymbol{\varphi}^{\mathcal{S}} \leftarrow \boldsymbol{\varphi}$ and $\boldsymbol{\mu}_{\boldsymbol{\varphi}}^{\mathcal{S}} \leftarrow \frac{1}{|\mathcal{D}|} \sum_{i=1}^{n} \nabla_{\boldsymbol{\varphi}} \mathcal{L}_i^{\boldsymbol{\varphi}}(\boldsymbol{\theta}, \boldsymbol{\varphi}^{\mathcal{S}})$
9:     $\boldsymbol{\theta}^{\mathcal{S}} \leftarrow \boldsymbol{\theta}$ and $\boldsymbol{\mu}_{\boldsymbol{\theta}}^{\mathcal{S}} \leftarrow \frac{1}{|\mathcal{Z}|} \sum_{i=1}^{n} \nabla_{\boldsymbol{\theta}} \mathcal{L}_i^{\boldsymbol{\theta}}(\boldsymbol{\theta}^{\mathcal{S}}, \boldsymbol{\varphi}^{\mathcal{S}})$
10:    $N \sim \mathrm{Geom}\left(1/n\right)$        $\triangleright$ Length of the epoch.
11:    **for** $i = 0$ **to** $N-1$ **do**
12:       **Sample** $i_{\boldsymbol{\theta}} \sim \pi_{\boldsymbol{\theta}}, i_{\boldsymbol{\varphi}} \sim \pi_{\boldsymbol{\varphi}}$, do **extrapolation:**
13:       $\tilde{\boldsymbol{\varphi}} \leftarrow \boldsymbol{\varphi} - \eta_{\boldsymbol{\theta}} \boldsymbol{d}_{\boldsymbol{\varphi}}(\boldsymbol{\theta}, \boldsymbol{\varphi}, \boldsymbol{\theta}^{\mathcal{S}}, \boldsymbol{\varphi}^{\mathcal{S}}) \ , \ \tilde{\boldsymbol{\theta}} \leftarrow \boldsymbol{\theta} - \eta_{\boldsymbol{\varphi}} \boldsymbol{d}_{\boldsymbol{\theta}}(\boldsymbol{\theta}, \boldsymbol{\varphi}, \boldsymbol{\theta}^{\mathcal{S}}, \boldsymbol{\varphi}^{\mathcal{S}})$    $\triangleright$ equation 9
       and equation 10
14:       **Sample** $i_{\boldsymbol{\theta}} \sim \pi_{\boldsymbol{\theta}}, i_{\boldsymbol{\varphi}} \sim \pi_{\boldsymbol{\varphi}}$, do **update:**
15:       $\boldsymbol{\varphi} \leftarrow \boldsymbol{\varphi} - \eta_{\boldsymbol{\theta}} \boldsymbol{d}_{\boldsymbol{\varphi}}(\tilde{\boldsymbol{\theta}}, \tilde{\boldsymbol{\varphi}}, \boldsymbol{\theta}^{\mathcal{S}}, \boldsymbol{\varphi}^{\mathcal{S}}) \ , \ \boldsymbol{\theta} \leftarrow \boldsymbol{\theta} - \eta_{\boldsymbol{\varphi}} \boldsymbol{d}_{\boldsymbol{\theta}}(\tilde{\boldsymbol{\theta}}, \tilde{\boldsymbol{\varphi}}, \boldsymbol{\theta}^{\mathcal{S}}, \boldsymbol{\varphi}^{\mathcal{S}})$    $\triangleright$ equation 9
       and equation 10
16:       $\bar{\boldsymbol{\theta}} \leftarrow \frac{t}{t+1} \bar{\boldsymbol{\theta}} + \frac{1}{t+1} \boldsymbol{\theta}$ and $\bar{\boldsymbol{\varphi}} \leftarrow \frac{t}{t+1} \bar{\boldsymbol{\varphi}} + \frac{1}{t+1} \boldsymbol{\varphi}$    $\triangleright$ Online computation of the average.
17:       $t \leftarrow t + 1$        $\triangleright$ Increment $t$ for the online average computation.
18:    **end for**
19: **end for**
20: **Output:** $\boldsymbol{\theta}, \boldsymbol{\varphi}$

---

### D.2    Hyperparameters used for the full-batch setting

**Optimal $\alpha$.** In the full-batch bilinear problem, it is possible to derive the optimal $\alpha$ parameter for a small enough $\eta$. Given the optimum $\boldsymbol{\omega}^{\star}$, the current iterate $\boldsymbol{\omega}$, and the "previous" iterate $\boldsymbol{\omega}^{\mathcal{P}}$ before $k$ steps, let $\boldsymbol{x} = \boldsymbol{\omega}^{\mathcal{P}} + \alpha(\boldsymbol{\omega} - \boldsymbol{\omega}^{\mathcal{P}})$ be the next iterate selected to be on the interpolated line between $\boldsymbol{\omega}^{\mathcal{P}}$ and $\boldsymbol{\omega}$. We aim at finding $\boldsymbol{x}$ (or in effect $\alpha$) that is closest to $\boldsymbol{\omega}^{\star}$. For an infinitesimally small learning rate, a GDA iterate would revolve around $\boldsymbol{\omega}^{\star}$, hence $\|\boldsymbol{\omega} - \boldsymbol{\omega}^{\star}\| = \|\boldsymbol{\omega}^{\mathcal{P}} - \boldsymbol{\omega}^{\star}\| = r$. The shortest distance between $\boldsymbol{x}$ and $\boldsymbol{\omega}^{\star}$ would be according to:

$$r^2 = \|\boldsymbol{\omega}^{\mathcal{P}} - \boldsymbol{x}\|^2 + \|\boldsymbol{x} - \boldsymbol{\omega}^{\star}\|^2 = \|\boldsymbol{\omega} - \boldsymbol{x}\|^2 + \|\boldsymbol{x} - \boldsymbol{\omega}^{\star}\|^2$$

Hence the optimal $\boldsymbol{x}$, for any $k$, would be obtained for $\|\boldsymbol{\omega} - \boldsymbol{x}\| = \|\boldsymbol{\omega}^{\mathcal{P}} - \boldsymbol{x}\|$, which is given for $\alpha = 0.5$.

In the case of larger learning rate, for which the GDA iterates diverge, we would have $\|\boldsymbol{\omega}^{\mathcal{P}} - \boldsymbol{\omega}^{\star}\| = r_1 < \|\boldsymbol{\omega} - \boldsymbol{\omega}^{\star}\| = r_2$ as we are diverging. Hence the optimal $\boldsymbol{x}$ would follow $\|\boldsymbol{\omega}^{\mathcal{P}} - \boldsymbol{x}\| < \|\boldsymbol{\omega} - \boldsymbol{x}\|$, which is given for $\alpha < 0.5$. In Fig. 3 we indeed observe LA-GDA with $\alpha = 0.4$ converging faster than with $\alpha = 0.5$.

**Hyperparameters.** Unless otherwise specified the learning rate used is fixed to $\eta = 0.3$. For both Unroll-Y and Unroll-XY, we use 6 unrolling steps. When combining Lookahead-minmax with GDA or Extragradient, we use a $k$ of 6 and $\alpha$ of 0.5 unless otherwise emphasized.

### D.3    Performance of EMA-iterates in the stochastic setting

For the identical experiment presented in § 4.2.2 in this section we depict the corresponding performance of the EMA iterates, of Adam, Extra-Adam, Extragradient and LA-GDA. The hyperparameters used are the same and shown in § D.4, we use $\beta = 0.999$ for EMA. In Fig. 7 we report the distance to the optimum as a function of the number of passes for each method. We observe the high variance of the stochastic bilinear is preventing the convergence of Adam, Extra-Adam and Extragradient,

when the batch size $B$ is small, despite computing an exponential moving average of the iterates over 20k iterations.

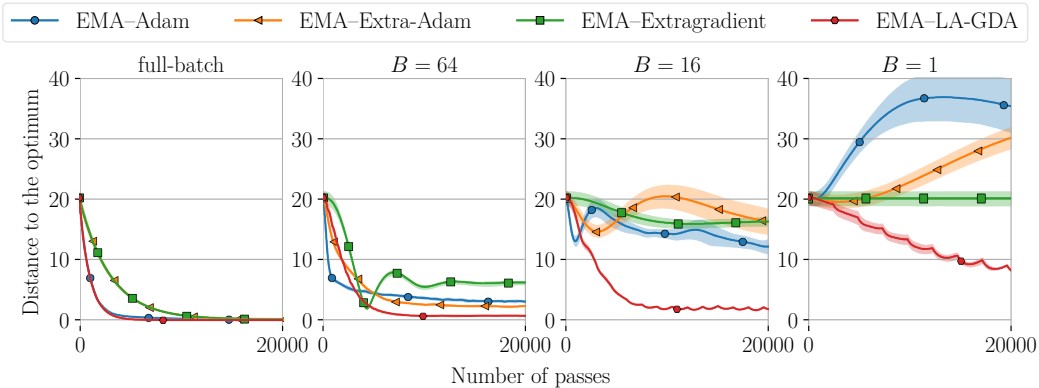

Figure 7: Distance to the optimum as a function of the number of passes, for Adam, Extra-Adam, Extragradient, and LA-GAN, all combined with EMA ($\beta = 0.999$). For small batch sizes, the high variance of the problem is preventing convergence for all methods but Lookahead-Minmax.

### D.4 HYPERPARAMETERS USED FOR THE STOCHASTIC SETTING

The hyperparameters used in the stochastic bilinear experiment of equation SB–G are listed in Table 4. We tuned the hyperparameters of each method independently, for each batch-size. We tried $\eta$ ranging from 0.005 to 1. When for all values of $\eta$ the method diverges, we set $\eta = 0.005$ in Fig. 4. To tune the first moment estimate of Adam $\beta_1$, we consider values ranging from $-1$ to 1, as Gidel et al. reported that negative $\beta_1$ can help in practice. We used $\alpha \in \{0.3, 0.5\}$ and $k \in [5, 3000]$.

| Batch-size | Parameter | Adam | Extra-Adam | Extragradient | LA-GDA | SVRE |
|---|---|---|---|---|---|---|
| full-batch | $\eta$ | 0.005 | 0.02 | 0.8 | 0.2 | - |
| | Adam $\beta_1$ | $-0.9$ | $-0.6$ | - | - | - |
| | Lookahead $k$ | - | - | - | 15 | - |
| | Lookahead $\alpha$ | - | - | - | 0.3 | - |
| 64 | $\eta$ | 0.005 | 0.01 | 0.005 | 0.005 | - |
| | Adam $\beta_1$ | $-0.6$ | $-0.2$ | - | - | - |
| | Lookahead $k$ | - | - | - | 450 | - |
| | Lookahead $\alpha$ | - | - | - | 0.3 | - |
| 16 | $\eta$ | 0.005 | 0.005 | 0.005 | 0.01 | - |
| | Adam $\beta_1$ | $-0.3$ | 0.0 | - | - | - |
| | Lookahead $k$ | - | - | - | 1500 | - |
| | Lookahead $\alpha$ | - | - | - | 0.3 | - |
| 1 | $\eta$ | 0.005 | 0.005 | 0.005 | 0.05 | 0.1 |
| | Adam $\beta_1$ | 0.0 | 0.0 | - | - | - |
| | Lookahead $k$ | - | - | - | 2450 | - |
| | Lookahead $\alpha$ | - | - | - | 0.3 | - |
| | restart probability $p$ | - | - | - | - | 0.1 |

Table 4: List of hyperparameters used in Figure 4. $\eta$ denotes the learning rate, $\beta_1$ is defined in equation 5, and $\alpha$ and $k$ in Alg. 1.

Fig. 8 depicts the final performance of Lookahead–minmax, using different values of $k$. Note that, the choice of plotting the distance to the optimum at a particular final iteration is causing the frequent oscillations of the depicted performances, since the iterate gets closer to the optimum only after the "backtracking" step. Besides the misleading oscillations, one can notice the trend of how the choice of $k$ affects the final distance to the optimum. Interestingly, the case of $B = 16$ in Fig. 8 captures the

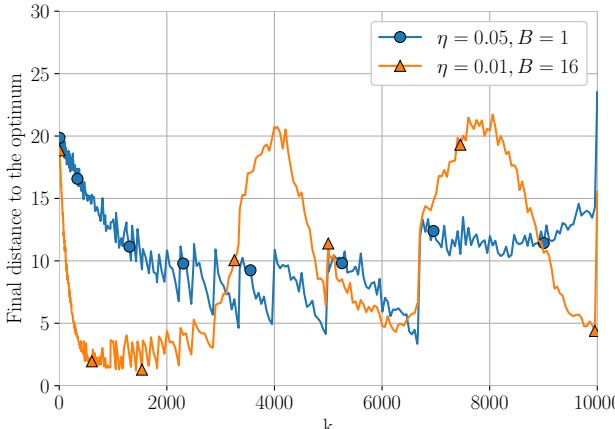

Figure 8: Sensitivity of LA-GDA to the value of the hyperparameter $k$ in Alg. 1 for two combinations of batch sizes and $\eta$. The y-axis is the distance to the optimum at 20000 passes. The jolting of the curves is due to the final value being affected by how close it is to the last equation LA step, *i.e.* lines 10 and 11 of Alg. 1.

periodicity of the rotating vector field, what sheds light on future directions in finding methods with adaptive $k$.

### D.5  NUMBER OF PASSES

In our toy experiments, as x-axis we use the "number of passes" as in (Chavdarova et al., 2019), so as to account for the different computation complexity of the optimization methods being compared. The *number of passes* is different from *the number of iterations*, where the former denotes one cycle of forward and backward passes, and alternatively, it can be called "gradient queries". More precisely, using parameter updates as the x-axis could be misleading as for example extragradient uses extra passes (gradient queries) per one parameter update. Number of passes is thus a good indicator of the computation complexity, as wall-clock time measurements–which depend on the concurrent overhead of the machine at the time of running, can be relatively more noisy.

### D.6  ILLUSTRATIONS OF GAN OPTIMIZATION WITH LOOKAHEAD

In Fig. 9 we consider a 2D bilinear game $\min_x \max_y x \cdot y$, and we illustrate the convergence of Lookahead–Minmax. Interestingly, Lookahead makes use of the rotations of the game vector field caused by the adversarial component of the game. Although standard-GDA diverges with all three shown learning rates, Lookahead–minmax converges. Moreover, we see Lookahead–minmax with larger learning rate of $\eta = 0.4$ (and fixed $k$ and $\alpha$) in fact converges faster then the case $\eta = 0.1$, what indicates that Lookahead–minmax is also sensitive to the value of $\eta$, besides that it introduces additional hyperparameters $k$ and $\alpha$

### E  EXPERIMENTS ON QUADRATIC FUNCTIONS

In this section we run experiments on the following 2D quadratic zero-sum problems:

$$\mathcal{L}(x, y) = -3x^2 + 4xy - y^2 \tag{QP-1}$$

$$\mathcal{L}(x, y) = x^2 + 5xy - y^2 \tag{QP-2}$$

### E.1  EXPERIMENTS ON EQUATION QP-1

In Fig. 10, we compare GDA, LA-GDA, ExtraGrad, and LA-ExtraGrad on the problem defined by equation QP-1. One can show that the spectral radius for the Jacobian of the GDA operator is always

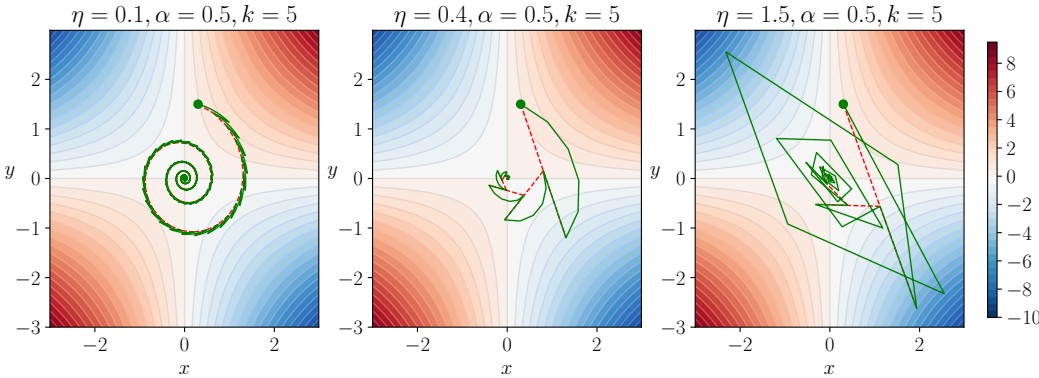

Figure 9: Illustration of Lookahead-minmax on the bilinear game $\min_x \max_y x \cdot y$, for different values of the learning rate $\eta \in \{0.1, 0.4, 1.5\}$, with fixed $k = 5$ and $\alpha = 0.5$. The trajectory of the iterates is depicted with green line, whereas the the interpolated line between $(\boldsymbol{\omega}_t, \tilde{\boldsymbol{\omega}}_{t,k})$, $t = 1, \ldots, T$, $k \in \mathbb{R}$ with $\boldsymbol{\omega}_t = (\boldsymbol{\theta}_t, \boldsymbol{\varphi}_t)$ is shown with dashed red line. The transparent lines depict the level curves of the loss function, and $\boldsymbol{\omega}^\star = (0.0)$. See § D.6 for discussion.

larger than one when considering a single step size $\eta$ shared by both players $x$ and $y$. Therefore, we tune each method by performing a grid search over individual step sizes for each player $\eta_x$ and $\eta_y$. We pick the pair $(\eta_x, \eta_y)$ which gives the smallest spectral radius. From Fig. 10 we observe that Lookahead-Minmax provides good performances.

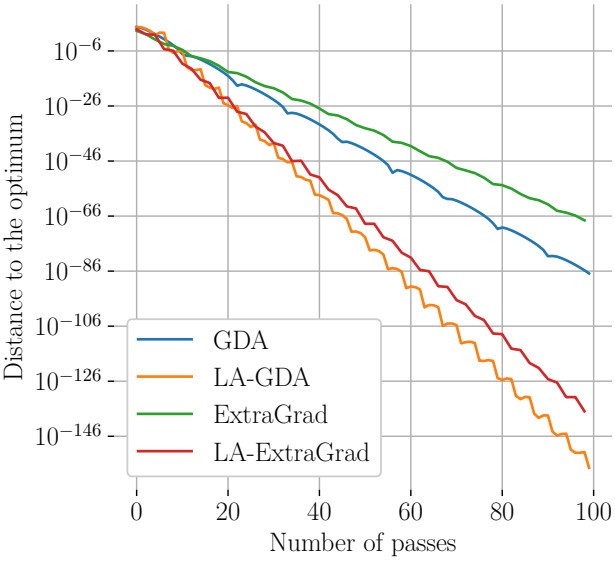

Figure 10: Convergence of GDA, LA-GDA, ExtraGrad, and LA-ExtraGrad on equation QP-1.

### E.2 EXPERIMENTS ON EQUATION QP-2

In Fig. 11, we compare GDA, LA-GDA, ExtraGrad, and LA-ExtraGrad on the problem defined by equation QP-2. This problem is better conditioned than equation QP-1 and one can show the spectral radius for the Jacobian of GDA and EG are smaller than one , for some step size $\eta$ shared by both players. In order to find the best hyperparameters for each method, we perform a grid search on the different hyperparameters and pick the set giving the smallest spectral radius. The results are corroborating our previous analysis as we observe a faster convergence when using Lookahead-Minmax.

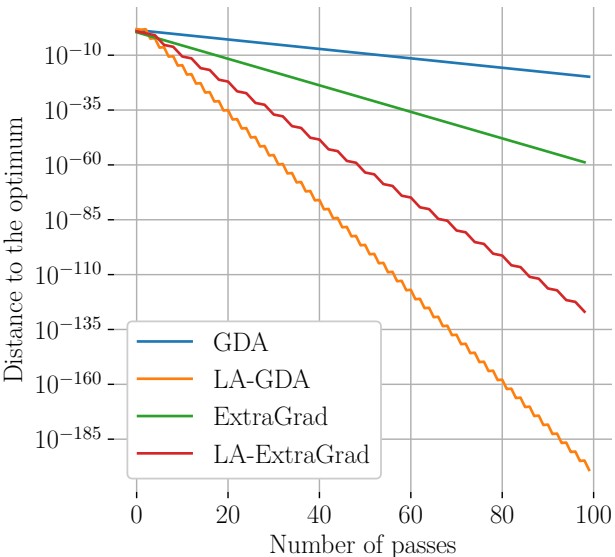

Figure 11: Convergence of GDA, LA-GDA, ExtraGrad, and LA-ExtraGrad on equation QP-2.

## F  PARAMETER AVERAGING

Polyak parameter averaging was shown to give fastest convergence rates among all stochastic gradient algorithms for convex functions, by minimizing the asymptotic variance induced by the algorithm (Polyak & Juditsky, 1992). This, so called *Ruppet–Polyak* averaging, is computed as the arithmetic average of the parameters:

$$\tilde{\boldsymbol{\theta}}_{RP} = \frac{1}{T} \sum_{t=1}^{T} \boldsymbol{\theta}^{(t)}, \quad T \geq 1 \,. \tag{RP–Avg}$$

In the context of games, weighted averaging was proposed by Bruck (1977) as follows:

$$\tilde{\boldsymbol{\theta}}_{\text{WA}}^{(T)} = \frac{\sum_{t=1}^{T} \rho^{(t)} \boldsymbol{\theta}^{(t)}}{\sum_{t=1}^{T} \rho^{(t)}} \,. \tag{W–Avg}$$

Eq. equation W–Avg can be computed efficiently *online* as: $\boldsymbol{\theta}_{\text{WA}}^{(t)} = (1 - \gamma^{(t)}) \boldsymbol{\theta}_{\text{WA}}^{(t-1)} + \gamma^{(t)} \boldsymbol{\theta}^{(t)}$ with $\gamma \in [0, 1]$. With $\gamma = \frac{1}{t}$ we obtain the Uniform Moving Averages (UMA) whose performance is reported in our experiments in § 5 and is computed as follows:

$$\boldsymbol{\theta}_{\text{UMA}}^{t} = (1 - \frac{1}{t}) \boldsymbol{\theta}_{\text{UMA}}^{(t-1)} + \frac{1}{t} \boldsymbol{\theta}^{(t)}, \quad t = 1, \ldots, T \,. \tag{UMA}$$

Analogously, we compute the Exponential Moving Averages (EMA) in an online fashion using $\gamma = 1 - \beta < 1$, as follows:

$$\boldsymbol{\theta}_{\text{EMA}}^{t} = \beta \boldsymbol{\theta}_{\text{EMA}}^{(t-1)} + (1 - \beta) \boldsymbol{\theta}^{(t)}, \quad t = 1, \ldots, T \,. \tag{EMA}$$

In our experiments, following related works (Yazıcı et al., 2019; Gidel et al., 2019a; Chavdarova et al., 2019), we fix $\beta = 0.9999$ for computing equation EMA *on all the fast weights* and when computing it on the slow weights in the experiments with small $k$ (e.g. when $k = 5$). The remaining case of computing equation EMA *on the slow weights* of the experiments that use large $k$ produce only *few* slow weight iterates, thus using such large $\beta = 0.9999$ results in largely weighting the parameters at initialization. We used fixed $\beta = 0.8$ for those cases.

## G  DETAILS ON THE LOOKAHEAD–MINMAX ALGORITHM AND ALTERNATIVES

As Lookahead uses iterates generated by an inner optimizer we also refer it is a *meta-optimizer* or *wrapper*. In the main paper we provide a general pseudocode that can be applied to any optimization method.

Alg. 3 reformulates the general Alg. 1 when combined with GDA, by replacing the inner fast-weight loop by a modulo operation of $k$–to clearly demonstrate to the reader the negligible modification for LA–MM of the source code of the base-optimizer. Note that instead of the notation of fast weights $\tilde{\theta}, \tilde{\varphi}$ we use snapshot networks $\theta^{\mathcal{S}}, \varphi^{\mathcal{S}}$ (past iterates) to denote the slow weights, and the parameters $\theta, \varphi$ for the fast weights. Alg. 3 also covers the possibility of using different update ratio $r \in \mathbb{Z}$ for the two players. For a fair comparison, all our empirical LA–GAN results use the convention of Alg. 3, *i.e.* as one step we count one update of both players (rather than one update of the slow weights–as it is $t$ in Alg. 1).

---

**Algorithm 3** Alternative formulation of Lookahead–Minmax (**equivalent** to Alg. 1& GDA)

1: **Input:** Stopping time $T$, learning rates $\eta_{\theta}, \eta_{\varphi}$, initial weights $\theta$, $\varphi$, lookahead hyperparameters $k$ and $\alpha$, losses $\mathcal{L}^{\theta}, \mathcal{L}^{\varphi}$, update ratio $r$, real–data distribution $p_d$, noise–data distribution $p_z$.
2: $\theta^{\mathcal{S}}, \varphi^{\mathcal{S}} \leftarrow \theta, \varphi$                                                       *(store snapshots)*
3: **for** $t \in 1, \ldots, T$ **do**
4:     **for** $i \in 1, \ldots, r$ **do**
5:         $\boldsymbol{x} \sim p_d, \boldsymbol{z} \sim p_z$
6:         $\varphi = \varphi - \eta_{\varphi} \nabla_{\varphi} \mathcal{L}^{\varphi}(\theta, \varphi, \boldsymbol{x}, \boldsymbol{z})$                 *(update $\varphi$ r times)*
7:     **end for**
8:     $\boldsymbol{z} \sim p_z$
9:     $\theta = \theta - \eta_{\theta} \nabla_{\theta} \mathcal{L}^{\theta}(\theta, \varphi, \boldsymbol{z})$                       *(update $\theta$ once)*
10:    **if** $t\%k == 0$ **then**
11:       $\varphi = \varphi^{\mathcal{S}} + \alpha_{\varphi}(\varphi - \varphi^{\mathcal{S}})$      *(backtracking on interpolated line $\varphi^{\mathcal{S}}$, $\varphi$)*
12:       $\theta = \theta^{\mathcal{S}} + \alpha_{\theta}(\theta - \theta^{\mathcal{S}})$      *(backtracking on interpolated line $\theta^{\mathcal{S}}$, $\theta$)*
13:       $\theta^{\mathcal{S}}, \varphi^{\mathcal{S}} \leftarrow \theta, \varphi$                          *(update snapshots)*
14:    **end if**
15: **end for**
16: **Output:** $\theta^{\mathcal{S}}, \varphi^{\mathcal{S}}$

---

Alg. 5 shows in detail how Lookahead–minmax can be combined with *Adam* equation Adam. Alg. 4 shows in detail how Lookahead–minmax can be combined with Extragradient. By combining equation Adam with Alg. 4 (analogous to Alg. 5), one could implement the LA–ExtraGrad–Adam algorithm, used in our experiments on MNIST, CIFAR-10, SVHN and ImageNet. A basic pytorch implementation of Lookahead-Minmax, including computing EMA on the slow weights, is provided in Listing. 1.

```python
from collections import defaultdict
import torch
import copy

class Lookahead(torch.optim.Optimizer):

    def __init__(self, optimizer, alpha=0.5):
        self.optimizer = optimizer
        self.alpha = alpha
        self.param_groups = self.optimizer.param_groups
        self.state = defaultdict(dict)

    def lookahead_step(self):
        for group in self.param_groups:
            for fast in group["params"]:
                param_state = self.state[fast]
                if "slow_params" not in param_state:
                    param_state["slow_params"] = torch.zeros_like(fast.data)
                    param_state["slow_params"].copy_(fast.data)
                slow = param_state["slow_params"]
                # slow <- slow+alpha*(fast-slow)
                slow += (fast.data - slow) * self.alpha
                fast.data.copy_(slow)

    def step(self, closure=None):
        loss = self.optimizer.step(closure)
        return loss

def update_ema_gen(G, G_ema, beta_ema=0.9999):
    l_param = list(G.parameters())
    l_ema_param = list(G_ema.parameters())
    for i in range(len(l_param)):
        with torch.no_grad():
            l_ema_param[i].data.copy_(l_ema_param[i].data.mul(beta_ema)
                            .add(l_param[i].data.mul(1-beta_ema)))

def train(G, D, optimizerD, optimizerG, data_sampler, noise_sampler,
          iterations=100, lookahead_k=5, beta_ema=0.9999,
          lookahead_alpha=0.5):

    # Wrapping the optimizers with the lookahead optimizer
    optimizerD = Lookahead(optimizerD, alpha=lookahead_alpha)
    optimizerG = Lookahead(optimizerG, alpha=lookahead_alpha)

    G_ema = None

    for i in range(iterations):
        # Update discriminator and generator
        discriminator_step(D, optimizerD, data_sampler, noise_sampler)
        generator_step(G, optimizerG, noise_sampler)

        if (i+1) % lookahead_k == 0: # Joint lookahead update
            optimizerG.lookahead_step()
            optimizerD.lookahead_step()
            if G_ema is None:
                G_ema = copy.deepcopy(G)
            else:
                # Update EMA on the slow weights
                update_ema_gen(G, G_ema, beta_ema=beta_ema)

    return G_ema
```

Listing 1: Implementation of Lookahead-minmax in Pytorch. Also includes the computation of EMA on the slow weights.

**Unroll-GAN Vs. Lookahead–Minmax (LA–MM).** LAGAN differs from UnrolledGAN (and approximated UnrolledGAN which does not backprop through the unrolled steps) as it: (i) *does not fix one player* to update the other player k times.    (ii) at each step $t$, it does *not* take gradient at future iterate $t + k$ to apply this gradient at the current iterate $t$ (as extragradient does too).    (iii) contrary to UnrolledGAN, LA–MM uses point on a *line between two iterates*, closeness to which is controlled with parameter $\alpha$, hence deals with rotations in a different way. Note that LA–MM can be applied to UnrolledGAN, however the heavy computational cost associated to UnrolledGAN prevented us from running this experiment (see note in Table 3 for computational comparison).

---

**Algorithm 4** Lookahead–Minmax combined with Extragradient (**equivalent** to Alg. 1 & EG)

---

 1: **Input:** Stopping time $T$, learning rates $\eta_{\boldsymbol{\theta}}, \eta_{\boldsymbol{\varphi}}$, initial weights $\boldsymbol{\theta}, \boldsymbol{\varphi}$, lookahead hyperparameters $k$ and $\alpha$, losses $\mathcal{L}^{\boldsymbol{\theta}}, \mathcal{L}^{\boldsymbol{\varphi}}$, update ratio $r$, real–data distribution $p_d$, noise–data distribution $p_z$.
 2: $\boldsymbol{\theta}^{\mathcal{S}}, \boldsymbol{\varphi}^{\mathcal{S}} \leftarrow \boldsymbol{\theta}, \boldsymbol{\varphi}$ *(store snapshots)*
 3: **for** $t \in 1, \ldots, T$ **do**
 4:    $\boldsymbol{\varphi}^{extra} \leftarrow \boldsymbol{\varphi}$
 5:    **for** $i \in 1, \ldots, r$ **do**
 6:       $\boldsymbol{x} \sim p_d, \boldsymbol{z} \sim p_z$
 7:       $\boldsymbol{\varphi}^{extra} = \boldsymbol{\varphi}^{extra} - \eta_{\boldsymbol{\varphi}} \nabla_{\boldsymbol{\varphi}^{extra}} \mathcal{L}^{\boldsymbol{\varphi}}(\boldsymbol{\theta}, \boldsymbol{\varphi}^{extra}, \boldsymbol{x}, \boldsymbol{z})$ *(Compute the extrapolated $\boldsymbol{\varphi}$)*
 8:    **end for**
 9:    $\boldsymbol{z} \sim p_z$
10:    $\boldsymbol{\theta}^{extra} = \boldsymbol{\theta} - \eta_{\boldsymbol{\theta}} \nabla_{\boldsymbol{\theta}} \mathcal{L}^{\boldsymbol{\theta}}(\boldsymbol{\theta}, \boldsymbol{\varphi}, \boldsymbol{z})$ *(Compute the extrapolated $\boldsymbol{\theta}$)*
11:    **for** $i \in 1, \ldots, r$ **do**
12:       $\boldsymbol{x} \sim p_d, \boldsymbol{z} \sim p_z$
13:       $\boldsymbol{\varphi} = \boldsymbol{\varphi} - \eta_{\boldsymbol{\varphi}} \nabla_{\boldsymbol{\varphi}} \mathcal{L}^{\boldsymbol{\varphi}}(\boldsymbol{\theta}^{extra}, \boldsymbol{\varphi}, \boldsymbol{x}, \boldsymbol{z})$ *(update $\boldsymbol{\varphi}$ $r$ times)*
14:    **end for**
15:    $\boldsymbol{z} \sim p_z$
16:    $\boldsymbol{\theta} = \boldsymbol{\theta} - \eta_{\boldsymbol{\theta}} \nabla_{\boldsymbol{\theta}} \mathcal{L}^{\boldsymbol{\theta}}(\boldsymbol{\theta}, \boldsymbol{\varphi}^{extra}, \boldsymbol{z})$ *(update $\boldsymbol{\theta}$ once)*
17:    **if** $t \% k == 0$ **then**
18:       $\boldsymbol{\varphi} = \boldsymbol{\varphi}^{\mathcal{S}} + \alpha_{\boldsymbol{\varphi}}(\boldsymbol{\varphi} - \boldsymbol{\varphi}^{\mathcal{S}})$ *(backtracking on interpolated line $\boldsymbol{\varphi}^{\mathcal{S}}$, $\boldsymbol{\varphi}$)*
19:       $\boldsymbol{\theta} = \boldsymbol{\theta}^{\mathcal{S}} + \alpha_{\boldsymbol{\theta}}(\boldsymbol{\theta} - \boldsymbol{\theta}^{\mathcal{S}})$ *(backtracking on interpolated line $\boldsymbol{\theta}^{\mathcal{S}}$, $\boldsymbol{\theta}$)*
20:       $\boldsymbol{\theta}^{\mathcal{S}}, \boldsymbol{\varphi}^{\mathcal{S}} \leftarrow \boldsymbol{\theta}, \boldsymbol{\varphi}$ *(update snapshots)*
21:    **end if**
22: **end for**
23: **Output:** $\boldsymbol{\theta}^{\mathcal{S}}, \boldsymbol{\varphi}^{\mathcal{S}}$

---

---

**Algorithm 5** Lookahead–Minmax (Alg. 1) combined with Adam as inner optimizer.

---

1: **Input:** Stopping time $T$, learning rates $\eta_\theta, \eta_\varphi$, initial weights $\theta$, $\varphi$, lookahead hyperparameters $k$ and $\alpha$, losses $\mathcal{L}^\theta$, $\mathcal{L}^\varphi$, update ratio $r$, real–data distribution $p_d$, noise–data distribution $p_z$, Adam parameters $\boldsymbol{m}^\varphi, \boldsymbol{v}^\varphi, \boldsymbol{m}^\theta, \boldsymbol{v}^\theta, \beta_1, \beta_2$.

2: $\theta^S, \varphi^S \leftarrow \theta, \varphi$                           *(store snapshots)*

3: $\boldsymbol{m}^\varphi, \boldsymbol{v}^\varphi, \boldsymbol{m}^\theta, \boldsymbol{v}^\theta \leftarrow \boldsymbol{0}, \boldsymbol{0}, \boldsymbol{0}, \boldsymbol{0}$       *(initialize first and second moments for Adam)*

4: **for** $t \in 1, \ldots, T$ **do**

5:     **for** $i \in 1, \ldots, r$ **do**

6:         $\boldsymbol{x} \sim p_d, \boldsymbol{z} \sim p_z$

7:         $\boldsymbol{g}^\varphi \leftarrow \nabla_\varphi \mathcal{L}^\varphi(\theta, \varphi, \boldsymbol{x}, \boldsymbol{z})$

8:         $\boldsymbol{m}^\varphi = \beta_1 \boldsymbol{m}^\varphi + (1 - \beta_1)\boldsymbol{g}^\varphi$

9:         $\boldsymbol{v}^\varphi = \beta_2 \boldsymbol{v}^\varphi + (1 - \beta_2)(\boldsymbol{g}^\varphi)^2$

10:        $\hat{\boldsymbol{m}}^\varphi = \frac{\boldsymbol{m}^\varphi}{1 - \beta_1^{((t-1)\times r + i)}}$

11:        $\hat{\boldsymbol{v}}^\varphi = \frac{\boldsymbol{v}^\varphi}{1 - \beta_2^{((t-1)\times r + i)}}$

12:        $\varphi = \varphi - \eta_\varphi \frac{\hat{\boldsymbol{m}}^\varphi}{\sqrt{\hat{\boldsymbol{v}}^\varphi} + \epsilon}$               *(update $\varphi$ $r$ times)*

13:     **end for**

14:     $\boldsymbol{z} \sim p_z$

15:     $\boldsymbol{g}^\theta \leftarrow \nabla_\theta \mathcal{L}^\theta(\theta, \varphi, \boldsymbol{x}, \boldsymbol{z})$

16:     $\boldsymbol{m}^\theta = \beta_1 \boldsymbol{m}^\theta + (1 - \beta_1)\boldsymbol{g}^\theta$

17:     $\boldsymbol{v}^\theta = \beta_2 \boldsymbol{v}^\theta + (1 - \beta_2)(\boldsymbol{g}^\theta)^2$

18:     $\hat{\boldsymbol{m}}^\theta = \frac{\boldsymbol{m}^\theta}{1 - \beta_1^t}$

19:     $\hat{\boldsymbol{v}}^\theta = \frac{\boldsymbol{v}^\theta}{1 - \beta_2^t}$

20:     $\theta = \theta - \eta_\theta \frac{\hat{\boldsymbol{m}}^\theta}{\sqrt{\hat{\boldsymbol{v}}^\theta} + \epsilon}$               *(update $\theta$ once)*

21:     **if** $t \% k == 0$ **then**

22:         $\varphi = \varphi^S + \alpha_\varphi(\varphi - \varphi^S)$     *(backtracking on interpolated line $\varphi^S$, $\varphi$)*

23:         $\theta = \theta^S + \alpha_\theta(\theta - \theta^S)$      *(backtracking on interpolated line $\theta^S$, $\theta$)*

24:         $\theta^S, \varphi^S \leftarrow \theta, \varphi$                     *(update snapshots)*

25:     **end if**

26: **end for**

27: **Output:** $\theta^S$, $\varphi^S$

---

### G.1 NESTED LOOKAHEAD–MINMAX

In our experiments we explored the effect of different values of $k$. On one hand, we observed that less stable baselines such as Alt-GAN tend to perform better using small $k$ (*e.g.* $k = 5$) when combined with Lookahead–Minmax, as it seems to be the necessary to prevent early divergence (what in turn achieves better iterate and EMA performances). On the other hand, we observed that stable baselines combined with Lookahead–Minmax with large $k$ (*e.g.* $k = 10000$) tend to take better advantage of the rotational behavior of games, as indicated by the notable improvement after each LA-step, see Fig. 2 and Fig. 12.

This motivates combination of both a small *"slow"* $k_s$ and a large *"super slow"* $k_{ss}$, as shown in Alg. 6. In this so called "nested" Lookahead–minmax version–denoted with *NLA* prefix, we store two copies of the each player, one corresponding to the traditional slow weights updated every $k_s$, and another for the so called *"super slow"* weights updated every $k_{ss}$. When computing EMA on the slow weights for NLA–GAN methods we use the super-slow weights as they correspond to the best performing iterates. However, better results could be obtain by computing EMA on the slow weights as EMA performs best with larger $\beta$ parameter and averaging over many iterates (whereas we obtain relatively small number of super–slow weights). We empirically find Nested Lookahead–minmax to be more stable than its non-nested counterpart, see Fig.13.

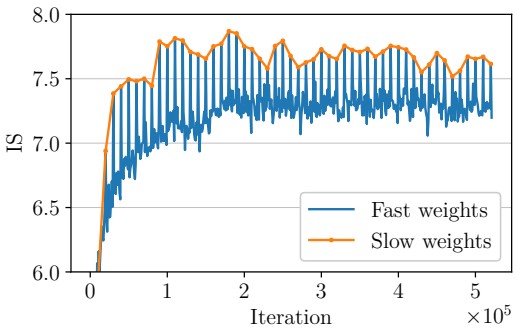

Figure 12: IS (higher is better) of LA–GAN on ImageNet with relatively large $k = 10000$. The backtracking step is significantly improving the model's performance every 10000 iterations. This shows how a large $k$ can take advantage of the rotating gradient vector field.

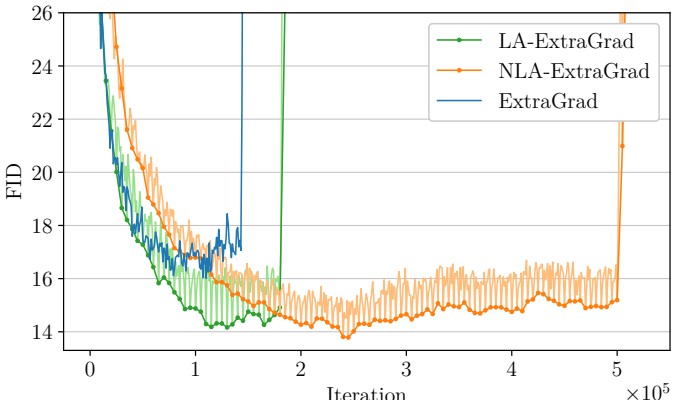

Figure 13: LA-ExtraGrad, NLA-ExtraGrad and Extragrad models trained on ImageNet. For LA-Extragrad ($k = 5000$), the lighter and darker colors represent the fast and slow weights respectively. For NLA-ExtraGrad ($k_s = 5, k_{ss} = 5000$), the lighter and darker colors represent the fast and super-slow weights respectively. In our experiments on ImageNet, while LA-ExtraGrad is more stable than ExtraGrad, it still has a tendency to diverge early. Using Alg. 6 we notice a clear improvement in stability.

## G.2 ALTERNATING–LOOKAHEAD–MINMAX

**Lookahead Vs. Lookahead–minmax.** Note how Alg. 1 differs from applying Lookahead to both the players *separately*. The obvious difference is for the case $r \neq 1$, as the backtracking is done at different number of updates of $\varphi$ and $\theta$. The key difference is in fact that after applying equation LA to one of the players, we do *not* use the resulting interpolated point to update the parameters of the other player–a version we refer to as "Alternating–Lookahead", see § G. Instead, equation LA is applied to both the players *at the same time*, which we found that outperforms the former. Unless otherwise emphasized, we focus on the "joint" version, as described in Alg. 1.

For completeness, in this section we consider an alternative implementation of Lookahead-minmax, which naively applies equation LA on each player separately, which we refer to as "alternating–lookahead". This in turn uses a "backtracked" iterate to update the opponent, rather than performing the "backtracking" step at the same time for both the players. In other words, the fact that line 9 of Alg. 7 is executed before updating $\theta$ in line 14, and vice versa, does not allow for Lookahead to help deal with the rotations typical for games.

---

**Algorithm 6** Pseudocode of Nested Lookahead–Minmax.

1: **Input:** Stopping time $T$, learning rates $\eta_{\boldsymbol{\theta}}, \eta_{\boldsymbol{\varphi}}$, initial weights $\boldsymbol{\theta}, \boldsymbol{\varphi}$, lookahead hyperparameters $k_s, k_{ss}$ and $\alpha$, losses $\mathcal{L}^{\boldsymbol{\theta}}, \mathcal{L}^{\boldsymbol{\varphi}}$, update ratio $r$, real–data distribution $p_d$, noise–data distribution $p_z$.

2: $(\boldsymbol{\theta}_s, \boldsymbol{\theta}_{ss}, \boldsymbol{\varphi}_s, \boldsymbol{\varphi}_{ss}) \leftarrow (\boldsymbol{\theta}, \boldsymbol{\theta}, \boldsymbol{\varphi}, \boldsymbol{\varphi})$         *(store copies for slow and super-slow)*
3: **for** $t \in 1, \ldots, T$ **do**
4:     **for** $i \in 1, \ldots, r$ **do**
5:         $\boldsymbol{x} \sim p_d, \boldsymbol{z} \sim p_z$
6:         $\boldsymbol{\varphi} \leftarrow \boldsymbol{\varphi} - \eta_{\boldsymbol{\varphi}} \nabla_{\boldsymbol{\varphi}} \mathcal{L}^{\boldsymbol{\varphi}}(\boldsymbol{\theta}, \boldsymbol{\varphi}, \boldsymbol{x}, \boldsymbol{z})$         *(update $\boldsymbol{\varphi}$ r times)*
7:     **end for**
8:     $\boldsymbol{z} \sim p_z$
9:     $\boldsymbol{\theta} \leftarrow \boldsymbol{\theta} - \eta_{\boldsymbol{\theta}} \nabla_{\boldsymbol{\theta}} \mathcal{L}^{\boldsymbol{\theta}}(\boldsymbol{\theta}, \boldsymbol{\varphi}, \boldsymbol{z})$         *(update $\boldsymbol{\theta}$ once)*
10:     **if** $t\%k_s == 0$ **then**
11:         $\boldsymbol{\varphi} \leftarrow \boldsymbol{\varphi}_s + \alpha_{\boldsymbol{\varphi}}(\boldsymbol{\varphi} - \boldsymbol{\varphi}_s)$         *(backtracking on interpolated line $\boldsymbol{\varphi}_s$, $\boldsymbol{\varphi}$)*
12:         $\boldsymbol{\theta} \leftarrow \boldsymbol{\theta}_s + \alpha_{\boldsymbol{\theta}}(\boldsymbol{\theta} - \boldsymbol{\theta}_s)$         *(backtracking on interpolated line $\boldsymbol{\theta}_s$, $\boldsymbol{\theta}$)*
13:         $(\boldsymbol{\theta}_s, \boldsymbol{\varphi}_s) \leftarrow (\boldsymbol{\theta}, \boldsymbol{\varphi})$         *(update slow checkpoints)*
14:     **end if**
15:     **if** $t\%k_{ss} == 0$ **then**
16:         $\boldsymbol{\varphi} \leftarrow \boldsymbol{\varphi}_{ss} + \alpha_{\boldsymbol{\varphi}}(\boldsymbol{\varphi} - \boldsymbol{\varphi}_{ss})$         *(backtracking on interpolated line $\boldsymbol{\varphi}_{ss}$, $\boldsymbol{\varphi}$)*
17:         $\boldsymbol{\theta} \leftarrow \boldsymbol{\theta}_{ss} + \alpha_{\boldsymbol{\theta}}(\boldsymbol{\theta} - \boldsymbol{\theta}_{ss})$         *(backtracking on interpolated line $\boldsymbol{\theta}_{ss}$, $\boldsymbol{\theta}$)*
18:         $(\boldsymbol{\theta}_{ss}, \boldsymbol{\varphi}_{ss}) \leftarrow (\boldsymbol{\theta}, \boldsymbol{\varphi})$         *(update super-slow checkpoints)*
19:         $(\boldsymbol{\theta}_s, \boldsymbol{\varphi}_s) \leftarrow (\boldsymbol{\theta}, \boldsymbol{\varphi})$         *(update slow checkpoints)*
20:     **end if**
21: **end for**
22: **Output:** $\boldsymbol{\theta}_{ss}, \boldsymbol{\varphi}_{ss}$

---

---

**Algorithm 7** Alternating Lookahead-minmax pseudocode.

1: **Input:** Stopping time $T$, learning rates $\eta_{\boldsymbol{\theta}}, \eta_{\boldsymbol{\varphi}}$, initial weights $\boldsymbol{\theta}, \boldsymbol{\varphi}, k_{\boldsymbol{\theta}}, k_{\boldsymbol{\varphi}}, \alpha_{\boldsymbol{\theta}}, \alpha_{\boldsymbol{\varphi}}$, losses $\mathcal{L}^{\boldsymbol{\theta}}$, $\mathcal{L}^{\boldsymbol{\varphi}}$, update ratio $r$, real–data distribution $p_d$, noise–data distribution $p_z$.
2: $\tilde{\boldsymbol{\theta}} \leftarrow \boldsymbol{\theta}$         *(store copy)*
3: $\tilde{\boldsymbol{\varphi}} \leftarrow \boldsymbol{\varphi}$
4: **for** $t \in 0, \ldots, T-1$ **do**
5:     **for** $i \in 1, \ldots, r$ **do**
6:         $\boldsymbol{x} \sim p_d, \boldsymbol{z} \sim p_z$
7:         $\boldsymbol{\varphi} \leftarrow \boldsymbol{\varphi} - \eta_{\boldsymbol{\varphi}} \nabla_{\boldsymbol{\varphi}} \mathcal{L}^{\boldsymbol{\varphi}}(\boldsymbol{\theta}, \boldsymbol{\varphi}, \boldsymbol{x}, \boldsymbol{z})$         *(update $\boldsymbol{\varphi}$ k times)*
8:         **if** $(t * r + i)\%k_{\boldsymbol{\varphi}} == 0$ **then**
9:             $\boldsymbol{\varphi} \leftarrow \tilde{\boldsymbol{\varphi}} + \alpha_{\boldsymbol{\varphi}}(\boldsymbol{\varphi} - \tilde{\boldsymbol{\varphi}})$         *(backtracking on line $\tilde{\boldsymbol{\varphi}}$, $\boldsymbol{\varphi}$)*
10:             $\tilde{\boldsymbol{\varphi}} \leftarrow \boldsymbol{\varphi}$
11:         **end if**
12:     **end for**
13:     $\boldsymbol{z} \sim p_z$
14:     $\boldsymbol{\theta} \leftarrow \boldsymbol{\theta} - \eta_{\boldsymbol{\theta}} \nabla_{\boldsymbol{\theta}} \mathcal{L}^{\boldsymbol{\theta}}(\boldsymbol{\theta}, \boldsymbol{\varphi}, \boldsymbol{z})$         *(update $\boldsymbol{\theta}$ once)*
15:     **if** $t\%k_{\boldsymbol{\theta}} == 0$ **then**
16:         $\boldsymbol{\theta} \leftarrow \tilde{\boldsymbol{\theta}} + \alpha_{\boldsymbol{\theta}}(\boldsymbol{\theta} - \tilde{\boldsymbol{\theta}})$         *(backtracking on line $\tilde{\boldsymbol{\theta}}$, $\boldsymbol{\theta}$)*
17:         $\tilde{\boldsymbol{\theta}} \leftarrow \boldsymbol{\theta}$
18:     **end if**
19: **end for**
20: **Output:** $\boldsymbol{\theta}, \boldsymbol{\varphi}$

---

On SVHN and CIFAR-10, the joint Lookahead-minmax consistently gave us the best results, as can be seen in Figure 14 and 15. On MNIST, the alternating and joint implementations worked equally well, see Figure 15.

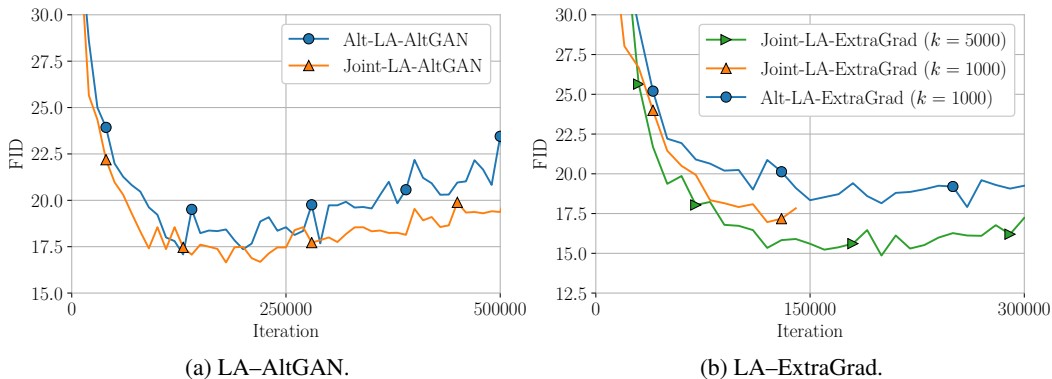

(a) LA–AltGAN.

(b) LA–ExtraGrad.

Figure 14: Comparison between different extensions of Lookahead to games on CIFAR-10. We use prefix *joint* and *alt* to denote Alg. 1 and Alg. 7, respectively of which the former is the one presented in the main paper. We can see some significant improvements in FID when using the joint implementation, for both LA-AltGAN (left) and LA-ExtraGrad (right).

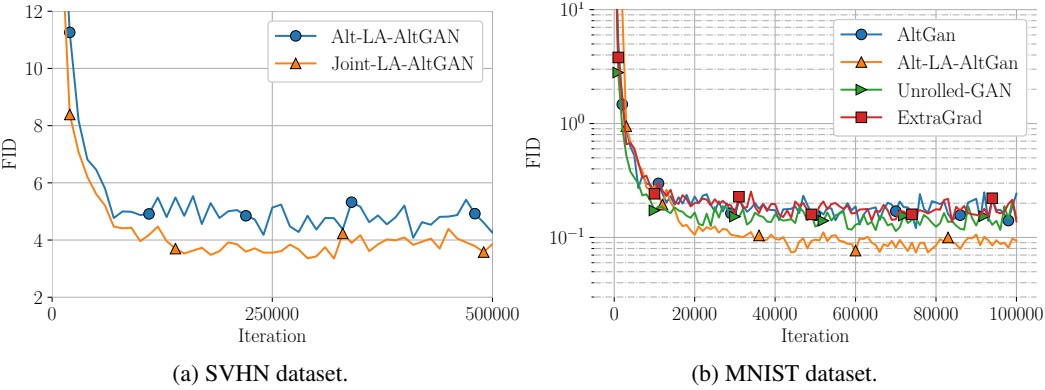

(a) SVHN dataset.

(b) MNIST dataset.

Figure 15: (a): Comparison of the joint Lookahead-minmax implementation (*Joint* prefix, see Algorithm 1) and the alternating Lookahead-minmax implementation (*Alt* prefix, see Algorithm 7) on the SVHN dataset. (b): Results obtained with the different methods introduced in §5 as well as an alternating implementation of Lookahead-minmax, on the MNIST dataset. Each curve is obtained averaged over 5 runs. The results of the alternating implementation differ very little from the joint implementation, the curve for Alt-LA-AltGAN matches results in Table 1.

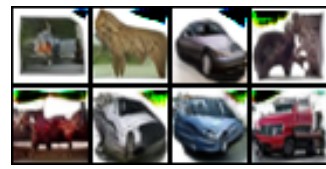

Figure 16: Samples from our generator model with the highest IS score. We can clearly see some unrealistic artefacts. We observed that the IS metric does not penalize these artefacts, whereas FID does penalize them.

# H    DETAILS ON THE IMPLEMENTATION

For our experiments, we used the PyTorch[2] deep learning framework. For experiments on CIFAR-10 and SVHN, we compute the FID and IS metrics using the provided implementations in Tensorflow[3] for consistency with related works. For experiments on ImageNet we use a faster-running PyTorch implementation[4] of FID and IS by (Brock et al., 2019) which allows for more frequent evaluation.

## H.1    METRICS

We provide more details about the metrics enumerated in § 5. Both FID and IS use: (i) the *Inception v3 network* (Szegedy et al., 2015) that has been trained on the ImageNet dataset consisting of ∼1 million RGB images of 1000 classes, $C = 1000$.   (ii) a sample of $m$ generated images $x \sim p_g$, where usually $m = 50000$.

### H.1.1    INCEPTION SCORE

Given an image $x$, IS uses the softmax output of the Inception network $p(y|x)$ which represents the probability that $x$ is of class $c_i, i \in 1 \dots C$, i.e., $p(y|x) \in [0,1]^C$. It then computes the marginal class distribution $p(y) = \int_x p(y|x)p_g(x)$. IS measures the Kullback–Leibler divergence $\mathbb{D}_{KL}$ between the predicted conditional label distribution $p(y|x)$ and the marginal class distribution $p(y)$. More precisely, it is computed as follows:

$$IS(G) = \exp\left(\mathbb{E}_{x \sim p_g}[\mathbb{D}_{KL}(p(y|x)||p(y))]\right) = \exp\left(\frac{1}{m}\sum_{i=1}^{m}\sum_{c=1}^{C}p(y_c|x_i)\log\frac{p(y_c|x_i)}{p(y_c)}\right). \quad (11)$$

It aims at estimating (i) if the samples look realistic i.e., $p(y|x)$ should have low entropy, and    (ii) if the samples are diverse (from different ImageNet classes) i.e., $p(y)$ should have high entropy. As these are combined using the Kullback–Leibler divergence, the higher the score is, the better the performance. Note that the range of IS scores at convergence varies across datasets, as the Inception network is pretrained on the ImageNet classes. For example, we obtain low IS values on the SVHN dataset as a large fraction of classes are numbers, which typically do not appear in the ImageNet dataset. Since **MNIST** has greyscale images, we used a classifier trained on this dataset and used $m = 5000$. For CIFAR-10 and SVHN, we used the original implementation[5] of IS in TensorFlow, and $m = 50000$.

As the Inception Score considers the classes as predicted by the Inception network, it can be prone not to penalize visual artefacts as long as those do not alter the predicted class distribution. In Fig. 16 we show some images generated by our best model according to IS. Those images exhibit some visible unrealistic artifacts, while enough of the image is left for us to recognise a potential image label. For this reason we consider that the Fréchet Inception Distance is a more reliable estimator of image quality. However, we reported IS for completeness.

---

[2]https://pytorch.org/
[3]https://www.tensorflow.org/
[4]https://github.com/ajbrock/BigGAN-PyTorch
[5]https://github.com/openai/improved-gan/

| Generator | Discriminator |
|---|---|
| *Input:* $z \in \mathbb{R}^{128} \sim \mathcal{N}(0, I)$ | *Input:* $x \in \mathbb{R}^{1 \times 28 \times 28}$ |
| transposed conv. (ker: 3×3, 128 → 512; stride: 1) | conv. (ker: 4×4, 1 → 64; stride: 2; pad:1) |
| Batch Normalization | LeakyReLU (negative slope: 0.2) |
| ReLU | conv. (ker: 4×4, 64 → 128; stride: 2; pad:1) |
| transposed conv. (ker: 4×4, 512 → 256, stride: 2) | Batch Normalization |
| Batch Normalization | LeakyReLU (negative slope: 0.2) |
| ReLU | conv. (ker: 4×4, 128 → 256; stride: 2; pad:1) |
| transposed conv. (ker: 4×4, 256 → 128, stride: 2) | Batch Normalization |
| Batch Normalization | LeakyReLU (negative slope: 0.2) |
| ReLU | conv. (ker: 3×3, 256 → 1; stride: 1) |
| transposed conv. (ker: 4×4, 128 → 1, stride: 2, pad: 1) | $Sigmoid(\cdot)$ |
| $Tanh(\cdot)$ | |

Table 5: DCGAN architectures (Radford et al., 2016) used for experiments on **MNIST**. We use *ker* and *pad* to denote *kernel* and *padding* for the (transposed) convolution layers, respectively. With $h \times w$ we denote the kernel size. With $c_{in} \rightarrow y_{out}$ we denote the number of channels of the input and output, for (transposed) convolution layers.

### H.1.2 FRÉCHET INCEPTION DISTANCE

Contrary to IS, FID aims at comparing the synthetic samples $x \sim p_g$ with those of the training dataset $x \sim p_d$ in a feature space. The samples are embedded using the first several layers of the Inception network. Assuming $p_g$ and $p_d$ are multivariate normal distributions, it then estimates the means $\boldsymbol{m}_g$ and $\boldsymbol{m}_d$ and covariances $C_g$ and $C_d$, respectively for $p_g$ and $p_d$ in that feature space. Finally, FID is computed as:

$$\mathbb{D}_{\text{FID}}(p_d, p_g) \approx d^2((\boldsymbol{m}_d, C_d), (\boldsymbol{m}_g, C_g)) = \|\boldsymbol{m}_d - \boldsymbol{m}_g\|_2^2 + Tr(C_d + C_g - 2(C_d C_g)^{\frac{1}{2}}), \quad (12)$$

where $d^2$ denotes the Fréchet Distance. Note that as this metric is a distance, the lower it is, the better the performance. We used the original implementation of FID[6] in Tensorflow, along with the provided statistics of the datasets.

### H.2 ARCHITECTURES & HYPERPARAMETERS

**Description of the architectures.** We describe the models we used in the empirical evaluation of Lookahead-minmax by listing the layers they consist of, as adopted in GAN works, e.g. (Miyato et al., 2018). With "conv." we denote a convolutional layer and "transposed conv" a transposed convolution layer (Radford et al., 2016). The models use Batch Normalization (Ioffe & Szegedy, 2015) and Spectral Normalization layers (Miyato et al., 2018).

### H.2.1 ARCHITECTURES FOR EXPERIMENTS ON MNIST

For experiments on the **MNIST** dataset, we used the DCGAN architectures (Radford et al., 2016), listed in Table 5, and the parameters of the models are initialized using PyTorch default initialization. For experiments on this dataset, we used the *non saturating* GAN loss as proposed (Goodfellow et al., 2014):

$$\mathcal{L}_D = \mathbb{E}_{x \sim p_d} \log(D(x)) + \mathbb{E}_{z \sim p_z} \log(D(G(z))) \quad (13)$$
$$\mathcal{L}_G = \mathbb{E}_{z \sim p_z} \log(D(G(z))), \quad (14)$$

where $p_d$ and $p_z$ denote the data and the latent distributions (the latter to be predefined).

### H.2.2 RESNET ARCHITECTURES FOR IMAGENET, CIFAR-10 AND SVHN

We replicate the experimental setup described for **CIFAR-10** and **SVHN** in (Miyato et al., 2018; Chavdarova et al., 2019), as listed in Table 7. This setup uses the hinge version of the adversarial non-saturating loss, see (Miyato et al., 2018). As a reference, our ResNet architectures for **CIFAR-10** have approximately 85 layers–in total for G and D, including the non linearity and the normalization layers.

---

[6] https://github.com/bioinf-jku/TTUR

| | **D–ResBlock ($\ell$–th block)** |
|---|---|
| | *Bypass*: |
| **G–ResBlock** | [AvgPool (ker:2×2 )], if $\ell = 1$ |
| | conv. (ker: 1×1, $3_{\ell=1}/128_{\ell\neq1} \to 128$; stride: 1) |
| *Bypass*: | Spectral Normalization |
| Upsample(×2) | [AvgPool (ker:2×2, stride:2)], if $\ell \neq 1$ |
| *Feedforward*: | *Feedforward*: |
| Batch Normalization | [ ReLU ], if $\ell \neq 1$ |
| ReLU | conv. (ker: 3×3, $3_{\ell=1}/128_{\ell\neq1} \to 128$; stride: 1; pad: 1) |
| Upsample(×2) | Spectral Normalization |
| conv. (ker: 3×3, $256 \to 256$; stride: 1; pad: 1) | ReLU |
| Batch Normalization | conv. (ker: 3×3, $128 \to 128$; stride: 1; pad: 1) |
| ReLU | Spectral Normalization |
| conv. (ker: 3×3, $256 \to 256$; stride: 1; pad: 1) | AvgPool (ker:2×2 ) |

Table 6: ResNet blocks used for the ResNet architectures (see Table 7), for the Generator (left) and the Discriminator (right). Each ResNet block contains skip connection (bypass), and a sequence of convolutional layers, normalization, and the ReLU non–linearity. The skip connection of the ResNet blocks for the Generator (left) upsamples the input using a factor of 2 (we use the default PyTorch upsampling algorithm–nearest neighbor), whose output is then added to the one obtained from the ResNet block listed above. For clarity we list the layers sequentially, however, note that the bypass layers operate in parallel with the layers denoted as "feedforward" (He et al., 2016). The ResNet block for the Discriminator (right) differs if it is the first block in the network (following the input to the Discriminator), $\ell = 1$, or a subsequent one, $\ell > 1$, so as to avoid performing the ReLU non–linearity immediate on the input.

| **Generator** | **Discriminator** |
|---|---|
| *Input: $z \in \mathbb{R}^{128} \sim \mathcal{N}(0, I)$* | *Input: $x \in \mathbb{R}^{3 \times 32 \times 32}$* |
| Linear($128 \to 4096$) | D–ResBlock |
| G–ResBlock | D–ResBlock |
| G–ResBlock | D–ResBlock |
| G–ResBlock | D–ResBlock |
| Batch Normalization | ReLU |
| ReLU | AvgPool (ker:8×8 ) |
| conv. (ker: 3×3, $256 \to 3$; stride: 1; pad:1) | Linear($128 \to 1$) |
| $Tanh(\cdot)$ | Spectral Normalization |

Table 7: *Deep* ResNet architectures used for experiments on **ImageNet**, **SVHN** and **CIFAR-10**, where G–ResBlock and D–ResBlock for the Generator (left) and the Discriminator (right), respectively, are described in Table 6. The models' parameters are initialized using the Xavier initialization (Glorot & Bengio, 2010). For ImageNet experiments, the generator's input is of dimension 512 instead of 128.

### H.2.3 UNROLLING IMPLEMENTATION

In Section D.1 we explained how we implemented unrolling for our full-batch bilinear experiments. Here we describe our implementation for our MNIST and CIFAR-10 experiments.

Unrolling is computationally intensive, which can become a problem for large architectures. The computation of $\nabla_\varphi \mathcal{L}^\varphi(\boldsymbol{\theta}_t^m, \boldsymbol{\varphi}_t)$, with $m$ unrolling steps, requires the computation of higher order derivatives which comes with a $\times m$ memory footprint and a significant slowdown. Due to limited memory, one can only backpropagate through the last unrolled step, bypassing the computation of higher order derivatives. We empirically see the gradient is small for those derivatives. In this approximate version, unrolling can be seen as of the same family as extragradient, computing its extrapolated points using more than a single step. We tested both true and approximate unrolling on MNIST, with a number of unrolling steps ranging from 5 to 20. The full unrolling that performs the backpropagation on the unrolled discriminator was implemented using the Higher[7] library. On CIFAR-10 we only experimented with approximate unrolling over 5 to 10 steps due to the large memory footprint of the ResNet architectures used for the generator and discriminator, making the other approach infeasible given our resources.

### H.2.4 HYPERPARAMETERS USED ON MNIST

Table 8 lists the hyperparameters that we used for our experiments on the MNIST dataset.

Table 8: Hyperparameters used on MNIST.

| Parameter | AltGAN | LA-AltGAN | ExtraGrad | LA-ExtraGrad | Unrolled-GAN |
|---|---|---|---|---|---|
| $\eta_G$ | 0.001 | 0.001 | 0.001 | 0.001 | 0.001 |
| $\eta_D$ | 0.001 | 0.001 | 0.001 | 0.001 | 0.001 |
| Adam $\beta_1$ | 0.05 | 0.05 | 0.05 | 0.05 | 0.05 |
| Batch-size | 50 | 50 | 50 | 50 | 50 |
| Update ratio $r$ | 1 | 1 | 1 | 1 | 1 |
| Lookahead $k$ | - | 1000 | - | 1000 | - |
| Lookahead $\alpha$ | - | 0.5 | - | 0.5 | - |
| Unrolling steps | - | - | - | - | 20 |

### H.2.5 HYPERPARAMETERS USED ON SVHN

Table 9: Hyperparameters used on SVHN.

| Parameter | AltGAN | LA-AltGAN | ExtraGrad | LA-ExtraGrad |
|---|---|---|---|---|
| $\eta_G$ | 0.0002 | 0.0002 | 0.0002 | 0.0002 |
| $\eta_D$ | 0.0002 | 0.0002 | 0.0002 | 0.0002 |
| Adam $\beta_1$ | 0.0 | 0.0 | 0.0 | 0.0 |
| Batch-size | 128 | 128 | 128 | 128 |
| Update ratio $r$ | 5 | 5 | 5 | 5 |
| Lookahead $k$ | - | 5 | - | 5000 |
| Lookahead $\alpha$ | - | 0.5 | - | 0.5 |

Table 9 lists the hyperparameters used for experiments on SVHN. These values were selected for each algorithm *independently* after tuning the hyperparameters for the baseline.

---

[7]https://github.com/facebookresearch/higher

### H.2.6 HYPERPARAMETERS USED ON CIFAR-10

Table 10: Hyperparameters that we used for our experiments on CIFAR-10.

| Parameter | AltGAN | LA-AltGAN | ExtraGrad | LA-ExtraGrad | Unrolled-GAN |
|---|---|---|---|---|---|
| $\eta_G$ | 0.0002 | 0.0002 | 0.0002 | 0.0002 | 0.0002 |
| $\eta_D$ | 0.0002 | 0.0002 | 0.0002 | 0.0002 | 0.0002 |
| Adam $\beta_1$ | 0.0 | 0.0 | 0.0 | 0.0 | 0.0 |
| Batch-size | 128 | 128 | 128 | 128 | 128 |
| Update ratio $r$ | 5 | 5 | 5 | 5 | 5 |
| Lookahead $k$ | - | 5 | - | 5000 | - |
| Lookahead $\alpha$ | - | 0.5 | - | 0.5 | - |
| Unrolling steps | - | - | - | - | 5 |

The reported results on CIFAR-10 were obtained using the hyperparameters listed in Table 10. These values were selected for each algorithm *independently* after tuning the hyperparameters. For the baseline methods we selected the hyperparameters giving the best performances. Consistent with the results reported by related works, we also observed that using larger ratio of updates of the discriminator and the generator improves the stability of the baseline, and we used $r = 5$. We observed that using learning rate decay delays the divergence, but does not improve the best FID scores, hence we did not use it in our reported models.

### H.2.7 HYPERPARAMETERS USED ON IMAGENEET

Table 11: Hyperparameters used for our AltGAN experiments on ImageNet.

| Parameter | AltGAN | LA-AltGAN | DLA-AltGAN |
|---|---|---|---|
| $\eta_G$ | 0.0002 | 0.0002 | 0.0002 |
| $\eta_D$ | 0.0002 | 0.0002 | 0.0002 |
| Adam $\beta_1$ | 0.0 | 0.0 | 0.0 |
| Batch-size | 256 | 256 | 256 |
| Update ratio $r$ | 5 | 5 | 5 |
| Lookahead $k$ | - | 5 | 5 |
| Lookahead $k'$ | - | - | 10000 |
| Lookahead $\alpha$ | - | 0.5 | 0.5 |
| Unrolling steps | - | - | - |

Table 12: Hyperparameters used for our ExtraGrad experiments on ImageNet.

| Parameter | ExtraGrad | LA-ExtraGrad | DLA-ExtraGrad |
|---|---|---|---|
| $\eta_G$ | 0.0002 | 0.0002 | 0.0002 |
| $\eta_D$ | 0.0002 | 0.0002 | 0.0002 |
| Adam $\beta_1$ | 0.0 | 0.0 | 0.0 |
| Batch-size | 256 | 256 | 256 |
| Update ratio $r$ | 5 | 5 | 5 |
| Lookahead $k$ | - | 5 | 5 |
| Lookahead $k'$ | - | - | 5000 |
| Lookahead $\alpha$ | - | 0.5 | 0.5 |
| Unrolling steps | - | - | - |

The reported results on ImageNet were obtained using the hyperparameters listed in Tables 11 and 12.

# I ADDITIONAL EXPERIMENTAL RESULTS

In Fig. 6 we compared the stability of LA–AltGAN methods against their AltGAN baselines on both the CIFAR-10 and SVHN datasets. Analogously, in Fig. 17 we report the comparison between LA–ExtraGrad and ExtraGrad over the iterations. We observe that the experiments on SVHN with ExtraGrad are more stable than those of CIFAR-10. Interestingly, we observe that: (i) LA–ExtraGradient improves both the stability and the performance of the baseline on CIFAR-10, see Fig. 17a, and (ii) when the stability of the baseline is relatively good as on the SVHN dataset, LA–Extragradient still improves its performances, see Fig. 17b.

For completeness to Fig. 5 in Fig 18 we report the respective eigenvalue analyses on MNIST, that are summarized in the main paper.

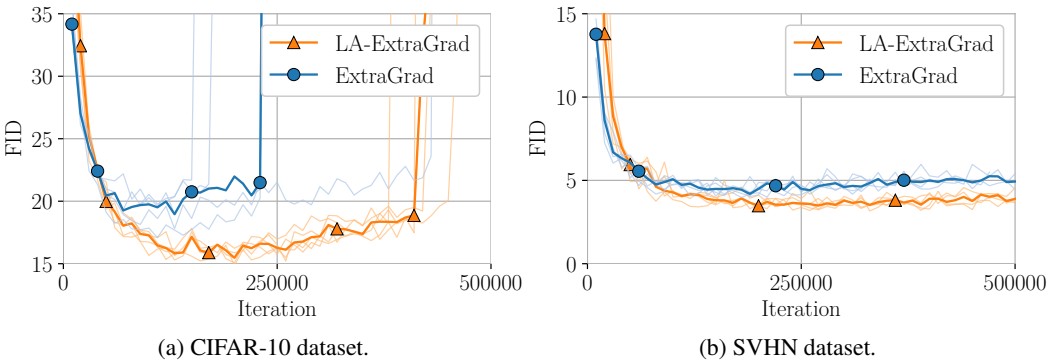

(a) CIFAR-10 dataset.  (b) SVHN dataset.

Figure 17: Improved stability of LA–ExtraGrad relative to its ExtraGrad baseline on SVHN and CIFAR-10, over 5 runs. The median and the individual runs are illustrated with ticker solid lines and with transparent lines, respectively. See § I and H for discussion and details on the implementation, resp.

## I.1 COMPLETE BASELINE COMPARISON

In § 5.2 we omitted uniform averaging of the iterates for clarity of the presented results–selected as it down-performs the exponential moving average in our experiments. In this section, for completeness we report the uniform averaging results. Table 13 lists these results, including experiments using the RAdam (Kingma & Ba, 2015) optimizer instead of Adam.

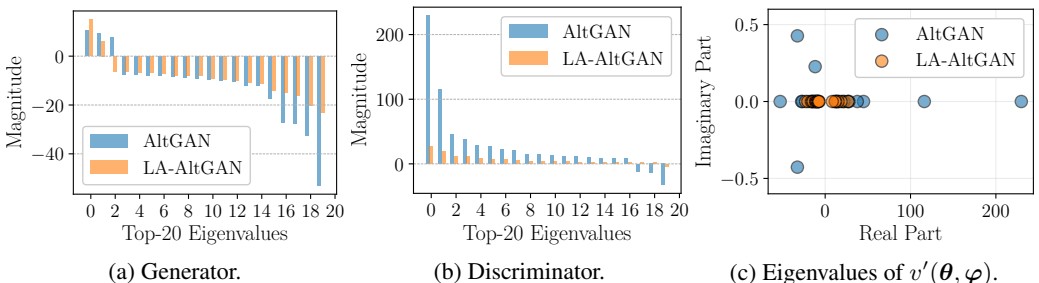

(a) Generator.  (b) Discriminator.  (c) Eigenvalues of $v'(\boldsymbol{\theta}, \boldsymbol{\varphi})$.

Figure 18: Analysis on MNIST at 100K iterations. Fig. 18a & 18b: Largest 20 eigenvalues of the Hessian of the generator and the discriminator. Fig. 18c: Eigenvalues of the Jacobian of JVF, indicating no rotations at the point of convergence of LA–AltGAN (see § 2).

| CIFAR-10 | Fréchet Inception distance | | | Inception score | | |
|---|---|---|---|---|---|---|
| Method | no avg | uniform avg | EMA | no avg | uniform avg | EMA |
| AltGAN–R | $23.27 \pm 1.65$ | $19.81 \pm 2.58$ | $17.82 \pm 1.31$ | $7.32. \pm .30$ | $8.14 \pm .20$ | $7.99 \pm .13$ |
| LA–AltGAN–R | $17.31 \pm .58$ | $16.68 \pm 1.45$ | $16.68 \pm 1.45$ | $7.81 \pm .09$ | $8.61 \pm .08$ | $8.13 \pm .07$ |
| ExtraGrad–R | $19.15 \pm 1.13$ | $16.17 \pm .63$ | $15.40 \pm .94$ | $7.59 \pm .13$ | $8.55 \pm .05$ | $8.05 \pm .15$ |
| LA–ExtraGrad–R | $15.38 \pm .76$ | $14.75 \pm .61$ | $14.99 \pm .66$ | $7.93 \pm .05$ | $8.51 \pm .08$ | $8.01 \pm .09$ |
| AltGAN–A | $21.37 \pm 1.60$ | $19.25 \pm 1.72$ | $16.92 \pm 1.16$ | $7.41 \pm .16$ | $8.23 \pm .17$ | $8.03 \pm .13$ |
| LA–AltGAN–A | $16.74 \pm .46$ | $15.02 \pm .81$ | $13.98 \pm .47$ | $8.05 \pm .43$ | $8.45 \pm .32$ | $8.19 \pm .05$ |
| ExtraGrad–A | $18.49 \pm .99$ | $16.22 \pm 1.59$ | $15.47 \pm 1.82$ | $7.61 \pm .07$ | $8.46 \pm .08$ | $8.05 \pm .09$ |
| LA–ExtraGrad–A | $15.25 \pm .30$ | $14.95 \pm .44$ | $14.68 \pm .30$ | $7.99 \pm .03$ | $8.13 \pm .18$ | $8.04 \pm .04$ |
| Unrolled–GAN–A | $21.04 \pm 1.08$ | $18.25 \pm 1.60$ | $17.51 \pm 1.08$ | $7.43 \pm .07$ | $8.26 \pm .15$ | $7.88 \pm .12$ |
| **SVHN** | | | | | | |
| AltGAN–A | $7.84 \pm 1.21$ | $10.83 \pm 3.20$ | $6.83 \pm 2.88$ | $3.10 \pm .09$ | $3.12 \pm .14$ | $3.19 \pm .09$ |
| LA–AltGAN–A | $3.87 \pm .09$ | $10.84 \pm 1.04$ | $3.28 \pm .09$ | $3.16 \pm .02$ | $3.38 \pm .09$ | $3.22 \pm .08$ |
| ExtraGrad–A | $4.08 \pm .11$ | $8.89 \pm 1.07$ | $3.22 \pm .09$ | $3.21 \pm .02$ | $3.21 \pm .04$ | $3.16 \pm .02$ |
| LA–ExtraGrad–A | $3.20 \pm .09$ | $7.66 \pm 1.54$ | $3.16 \pm .14$ | $3.20 \pm .02$ | $3.32 \pm .13$ | $3.19 \pm .03$ |
| **MNIST** | | | | | | |
| AltGAN–A | $.094 \pm .006$ | $.167 \pm .033$ | $.031 \pm .002$ | $8.92 \pm .01$ | $8.88 \pm .02$ | $8.99 \pm .01$ |
| LA–AltGAN–A | $.053 \pm .004$ | $.176 \pm .024$ | $.029 \pm .002$ | $8.93 \pm .01$ | $8.92 \pm .01$ | $8.96 \pm .02$ |
| ExtraGrad–A | $.094 \pm .013$ | $.182 \pm .024$ | $.032 \pm .003$ | $8.90 \pm .01$ | $8.88 \pm .03$ | $8.98 \pm .01$ |
| LA–ExtraGrad–A | $.053 \pm .005$ | $.180 \pm .024$ | $.032 \pm .002$ | $8.91 \pm .01$ | $8.92 \pm .02$ | $8.95 \pm .01$ |
| Unrolled–GAN–A | $.077 \pm .006$ | $.224 \pm .016$ | $.030 \pm .002$ | $8.91 \pm .02$ | $8.91 \pm .02$ | $8.99 \pm .01$ |

Table 13: Comparison of the LA-GAN optimizer with its respective baselines AltGAN and ExtraGrad (see § 5.1 for naming), using FID (lower is better) and IS (higher is better). EMA denotes *exponential moving average* (with fixed $\beta = 0.9999$, see § F). With suffix –R and –A we denote that we use *RAdam* (Liu et al., 2020) and *Adam* (Kingma & Ba, 2015) optimizer, respectively. Results are averaged over 5 runs. We run each experiment on MNIST for 100K iterations, and for 500K iterations for the rest of the datasets. See § H and § 5.2 for details on architectures and hyperparameters and for discussion on the results, resp.

## I.2 SAMPLES OF LA–GAN GENERATORS

In this section we show random samples of the generators of our LAGAN experiments trained on ImageNet, CIFAR-10 and SVHN.

The samples in Fig. 19 are generated by our best performing LA-AltGAN models trained on CIFAR-10. Similarly, Fig. 20 & 21 depict such samples of generators trained on ImageNet and SVHN, respectively.

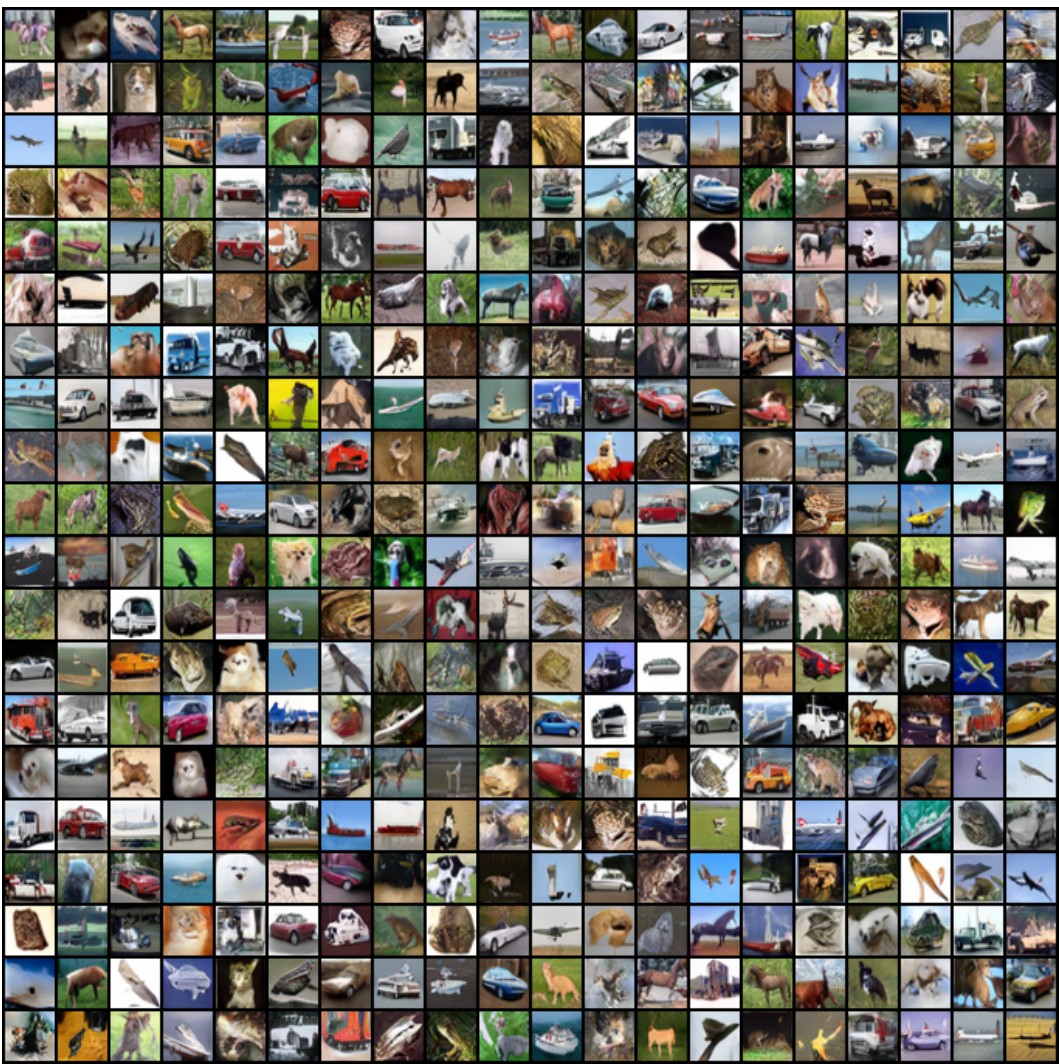

Figure 19: Samples generated by our best performing trained generator on **CIFAR-10**, using **LA-AltGAN** and exponential moving average (EMA) on the *slow* weights. The obtained FID score is 12.193.

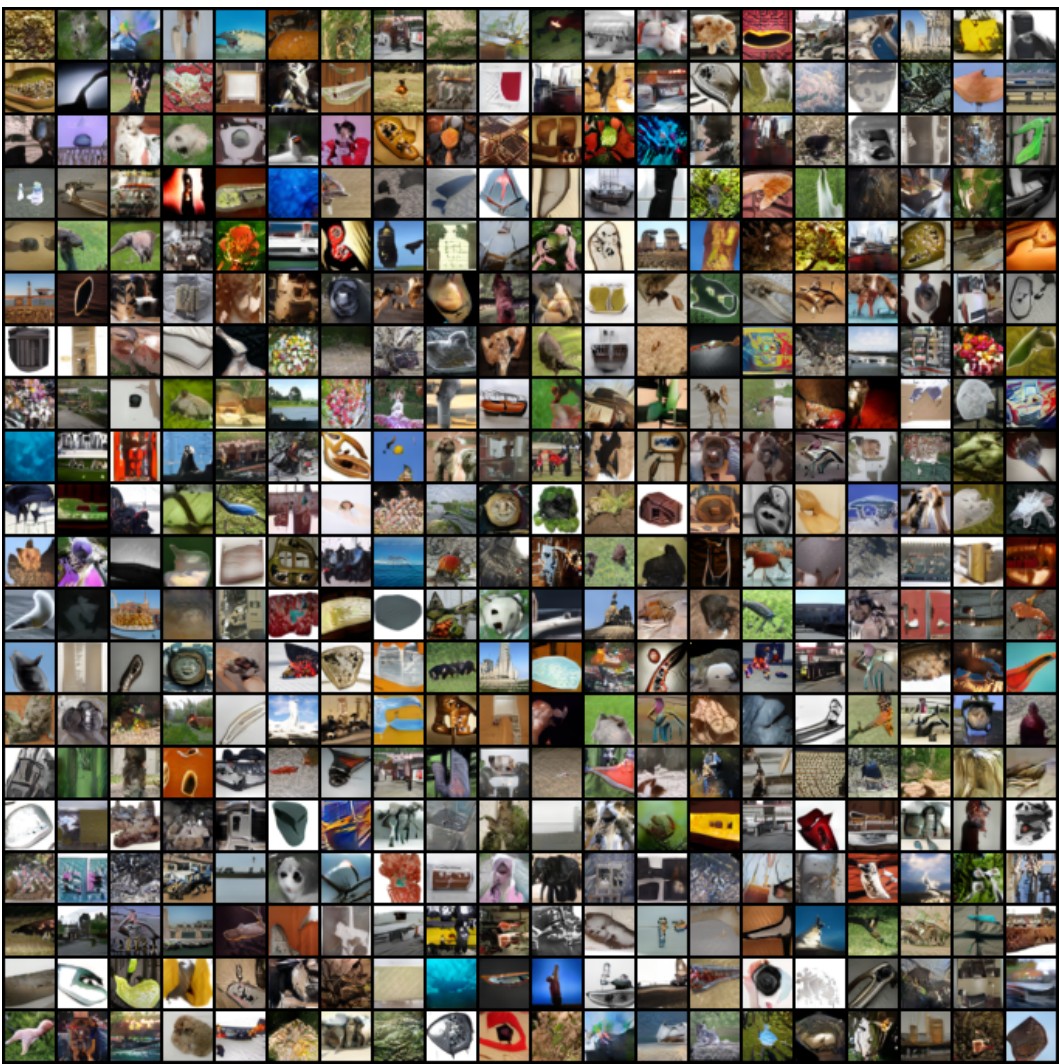

Figure 20: Images generated by our best model trained on **32×32 ImageNet**, obtained with **LA-AltGAN** and EMA of the *slow weights*, yielding FID of 12.44.

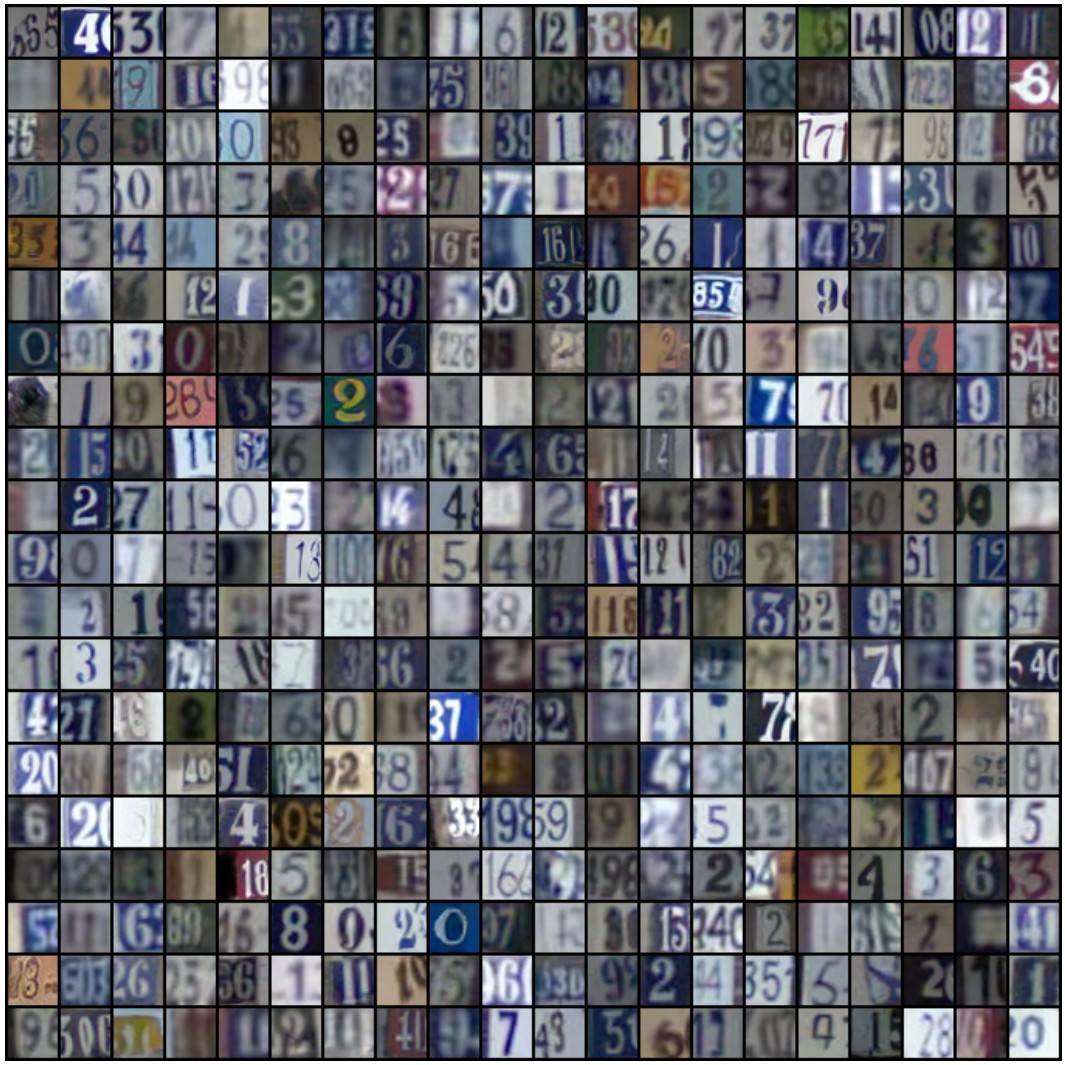

Figure 21: Images generated by one of our best **LA-ExtraGrad** & EMA model (FID of 2.94) trained on **SVHN**.

