# OpenReview forum: "Taming GANs with Lookahead-Minmax"
_ICLR.cc/2021/Conference — ICLR 2021 Poster_

### Official Review · AnonReviewer4 · 2020-10-20
**Strong empirical results on standard benchmarks, but no theoretical result provided.**

**Rating:** 7
**Confidence:** 5

**Review:**

Summary: This work extends the recently proposed lookahead optimizer (which was designed for single-objective optimization) to minimax optimization, particularly GAN training. The authors claim that the backtracking step in lookahead optimizer alleviates the notorious rotational behavior in GAN dynamics. Moreover, the authors argue that the lookahead optimizer implicitly handles the high variance in the small-batch setting. Both arguments are backed up by toy experiments on stochastic bilinear games.  Finally, on standard image datasets, the lookahead minimax algorithm outperforms some popular algorithms and achieves state-of-the-art performance on CIFAR-10.

Review: This paper is well-written and explains the main idea in a clear and effective manner. The empirical results of the introduced lookahead minimax algorithm are quite impressive given its simplicity. However, the lookahead minimax dynamics is not well-understood and no theoretical justification is provided in the paper. The bilinear game might not reflect the real GAN training as standard GDA never diverges in real GAN training. I tend to give a score of 7 or even higher, but I'm not satisfied with the current explanation of the performance gain. I think a better understanding of the dynamics is needed and I will increase my rating if the authors could improve the paper accordingly.

Here are detailed comments:

1. In the third paragraph of the introduction, the authors mentioned that empirically GANs often converge to a locally stable stationary point that is not a differential Nash equilibrium. Actually, this phenomenon occurs even for the simplest quadratic minimax games, see for example [1, 2, 3]. Moreover, Nash equilibrium might not even exist in general, Stackelberg equilibrium [2, 4] is probably a better solution notion.

2. The authors provide some theoretical results of the lookahead optimizer, but for single-objective optimization. I suggest the authors moving this section to the Appendix (as this section seems irrelevant) and replace it with the analysis of lookahead for minimax games. In particular, the authors argue many times in the paper that the lookahead algorithm is able to handle the rotational dynamics well, I think the authors should be able to verify that theoretically on quadratic minimax games if the argument is right.

3. The bilinear game might not reflect the real training dynamics of GANs. In bilinear games, simultaneous GDA with finite step sizes simply diverges, but it is not the case in practice for GAN training. To this end, strongly-convex strongly-concave minimax games might be better to model the underlying GAN dynamics. Strongly-convex strongly-concave quadratic minimax games shall be as simple as the bilinear cases and some theoretical analyses and simulation should be easy to do.

4. In the paper, the authors chose Extragradient as one of their baselines. However, after inspecting the code, I find much fewer iterations are used for Extragradient. I understand the authors would like to keep the total gradient queries the same over different algorithms, but there is a similar algorithm (Optimistic gradient descent ascent [5, 6]) with only one gradient query per step and perform similarly with Extragradient. In this sense, I think it is fair to only use fewer iterations for Extragradient.

5. For stochastic games (especially for SB-G), there is a better version of Extragradient which handles the variance well, see [7] for details.

6. In the original lookahead paper, it was mentioned that the slow weights can be understood as the exponential moving average (EMA) of the fast weights. In particular, the authors of the original lookahead paper did some analysis on a noisy quadratic model and showed lookahead reduces variance. In that simple noisy quadratic model, exponential moving average can effectively reduce variance, see section 3.4 of [8] for details. I think it is probably worth testing EMA on the toy experiments and mentioning it in the paper. I notice that the authors have already included EMA in section 5 and EMA improves the performance a lot. To this end, I think the advantages of lookahead can be potentially decomposed into two parts (though more works need to be done): (1) it suppresses the rotational part in game dynamics (I encourage the authors to analyze the spectrum of the dynamics); (2) it implicitly does exponential moving average and hence reduces the variance.

Overall, I think this paper is very strong on the empirical results. However, analysis on the dynamics of the lookahead algorithm is lacking. I will be happy to increase my rating if the authors could resolve some of my comments.

--------------------
**In the rebuttal, the authors resolved most of my concerns, therefore I increased my score to 7 as promised.**


Reference:
[1] On Finding Local Nash Equilibria (and Only Local Nash Equilibria) in Zero-Sum Games, 2019.

[2] On Solving Minimax Optimization Locally: A Follow-the-Ridge Approach, 2020.

[3] GANs May Have No Nash Equilibria, 2020.

[4] Convergence of learning dynamics in Stackelberg games, 2019.

[5] Training GANs with Optimism, 2018.

[6] A Unified Analysis of Extra-gradient and Optimistic Gradient Methods for Saddle Point Problems: Proximal Point Approach, 2019.

[7] Revisiting Stochastic Extragradient, 2019.

[8] Which Algorithmic Choices Matter at Which Batch Sizes? Insights From a Noisy Quadratic Model, 2019.

---

> ### Author Response · Authors · 2020-11-20
> **Thanks for the great review!**
>
> Many thanks for your extensive review and detailed comments. Below we answer separately to your detailed comments.
>
> 1. In our understanding, recent works give a game-theoretic interpretation of these points (as diff. Stackelberg Equilibria - DSE), however, as these are a superset of DNEs it remains unclear if the performance of the leader at DSE (on average) is as good as at DNEs (and approximate DNEs are guaranteed to exist [Daskalakis, Skoulakis, Zampetakis, Sep.2020]). We remain brief in this section by only mentioning the needed info (which we consider valid) to present our empirical result that LA converges to a point which has no rotation-Fig. 5 (relevant to show, as to our knowledge, such a case hasn’t been observed so far in GAN training). We added a new section in the supplementary (App. C, referenced from the 3rd paragraph of Sec. 2), where we also discuss the references you listed.
>
> 2. We aim to keep this paper mainly empirical for clarity, while still clarifying that besides that Lookahead-Minmax has been shown to work well in practice there is no result (to our knowledge) on improved upper bound on its convergence rate (which we consider important to clarify). Given the extra page allowed (relative to the initial submission), we’d like to keep this result in the main paper.
> To address your main concern--verifying theoretically our argument on ‘handles well rotations’, please see our revised section 4 and App. B. (also related to your last question 6), thanks!
>
> 3. Thanks for the suggestion, we considered a quadratic function from the FtR-Approach in reference [2], and we did a spectral analysis comparing Extragradient (EG), GDA, and their Lookahead-Minmax counterparts LA-EG, and LA-GDA. We computed the spectral radius of those operators for a wide range of pairs of learning rates for each player and saved the parameters giving us the best contraction. In figure 10, we plot the distance to the optimum as a function of the number of passes, for the best contraction. Our results are in line with our bilinear experiments, as (i) Lookahead-Minmax methods have the best contraction when compared to GDA and EG, and (ii) Lookahead-Minmax improves the contraction of the base optimizer.
> We think the bilinear game (BG) is relevant as it has high rotation--which standard minimization algorithms don’t handle well (and SB-G has both high rotation and noise), is widely studied,  and finally, we find it empirically relevant as: (i) although alt-GDA does not diverge at the beginning of the training (as on BG), it diverges notably earlier than EG)  (ii) we see empirical relevance as pointed out in Fig. 2.
>
> 4. As both EG and Optimistic Gradient Descent Ascent (OGDA) can be seen as approximating Proximal Point Method [Mokhtari et al. 2019] we selected one of these. We added OGDA in Fig. 3, where OGDA exploits well the fact that the gradient is smooth for the bilinear game (to approximate EG with past gradient), and outperforms EG when the x-axis is the gradient queries. For the larger-scale experiments, we selected EG and trained all methods for a fixed number of parameter updates (not gradient queries, so we think OGDA can perform as good or worse as the reported results for EG).
>
> [Mokhtari et al.] A Unified Analysis of Extra-gradient and Optimistic Gradient Methods for Saddle Point Problems: Proximal Point Approach, 2019
>
> 5. Thanks a lot for the reference, we’ll try to add it in our next revision.
>
> 6. We added the result of EMA on SB-G in the App. D.3. We do not show EMA for toy-experiments in the main paper, as for convex-concave objective EMA is less interesting than the uniform average (UA) (one can show that UA of GDA on a bounded domain converges due to convexity, which to our knowledge cannot be shown for EMA). On the other hand, UA is not interesting for DNNs, as in a highly non-convex non-concave DNN setting we can no longer rely on convexity for UA to converge, and EMA is shown to perform notably better than UA in practice (consistent with our UA results in App.), see discussions on ‘last iterate convergence’ in [1,2].
> Thanks a lot for your recommendation to analyze the spectrum of the dynamics! See our revised section 4 and  App. B.
>
>
> [1] Reducing Noise in GAN Training with Variance Reduced Extragradient, 2019
>
> [2] Stochastic Hamiltonian Gradient Methods for Smooth Games, 2020

---

> > ### Author Response · Authors · 2020-11-25
> > **Update on detailed comment 5**
> >
> > We’d like to note that [1] mentions the SB-G problem in the motivation, but does not compare the one-sample-EG on this problem empirically (the empirical evaluation contains a different bilinear game). We tested it on SB-G and noticed that taking the same sample for the extrapolation for the experiment in Fig.4: (i) does not make the diverging methods in Fig.4 converge, and (ii) for minibatch size 64, it speeds up slightly the otherwise converging Extra-Adam,  and it did not speed up Extra-Gradient.
> >
> > [1] Revisiting Stochastic Extragradient, 2019.
> >
> > [2] Reducing Noise in GAN Training with Variance Reduced Extragradient, 2019
> >
> > Please let us know if you have further comments, thanks!

---

### Official Review · AnonReviewer1 · 2020-10-26
**Official Blind Review #1**

**Rating:** 6
**Confidence:** 3

**Review:**

##### Summary:

This paper proposes a lookahead-minmax algorithm for optimizing minmax problems such as GANs, which updates the parameters (of both the generator and the discriminator) with the extrapolation. With a bilinear example, the authors show that the use of lookahead-minimax allows for convergence in cases where other methods does not, and yields good performance under high variance. Experiments of generative performance on several well-known public datasets demonstrates the effectiveness of the proposed method.

##### Reasons for score:

The idea is intuitive and easy to follow. My major concern is about the scale of the experiments.

---
##### Pros:

1. The paper is well-motivated and the idea is easy to understand. Figure 1 clearly demonstrates how lookahead-minmax can address the rotational nature in GAN’s training, which is further confirmed in Figure 5.

2. The advantages of the proposed lookahead-minmax is demonstrated with a bilinear example. For me, this helps to understand the merits of the proposed method.

3. The paper provides comprehensive experiments and describes the experimental settings in details.

##### Cons:

1. The contributions are a little bit confusing. The first contribution listed in the introduction (and Section 3) is an improved convergence guarantee, which seems an independent contribution. What is the relation between this theoretical result and the lookahead-minmax algorithm for GANs? Does the following analysis and conclusions rely on this improvement? After reading the paper, I found the rest of the paper does not necessarily build on the new convergence guarantee. I think it might be clearer to mention the most significant contribution first.

2. Some related works are missing. Gidel et al. 2019b presents the negative momentum method, which leads to a similar “backtracking step” in the vector field the gradients. CLC-GAN [Xu et al. arXiv:1909.13188] proposes to stabilize GANs’ training dynamics by interpreting the gradient flow as a dynamic system and manage it with closed-loop control, which is also related to this paper.

3. The experiments are conducted on 4 representative datasets. However, the image sizes (32*32) do not match the state-of-the-art GANs, which typically can generate larger images (e.g. Mescheder et al. 2018). One concern is whether the proposed method works well on larger datasets such as LSUN and ImageNet with higher resolutions. It would be more convincing if the authors can provide results on larger datasets.

---
##### Some typos:
(1) In Eqn. (JVF): the second entry in the JVF $\nabla_{\phi}\mathcal{L}^{\theta}$ -> $\nabla_{\phi}\mathcal{L}^{\phi}$

(2) In the line below Eqn. (SB-G): It seems that $b$ and $c$ are of dimension $d \times n$.

(3) In paragraph of “Benchmark on CIFAR-10 …” of Page 7, “3.5 times larger model then ours” -> “3.5 times larger model than ours”

(4) In the same paragraph mentioned in (3), the references to tables are inconsistent. Both “Table 2” and “Tab. 2” are used.

(5) In the last line of Page 7, “LA—GAN” em dash -> en dash

---

> ### Author Response · Authors · 2020-11-20
> **Thanks for your feedback and for pointing out our typos!**
>
> Thanks a lot for catching the typos!
> > The contributions are a little bit confusing. The first contribution listed in the introduction (and Section 3) is an improved convergence guarantee, which seems an independent contribution. What is the relation between this theoretical result and the lookahead-minmax algorithm for GANs? Does the following analysis and conclusions rely on this improvement? After reading the paper, I found the rest of the paper does not necessarily build on the new convergence guarantee. I think it might be clearer to mention the most significant contribution first.
>
> Thanks for pointing this out! We re-ordered that contribution as second, and we added to the same bullet point our theoretical insight on lookahead-minmax (added in Sec. 4 of our latest revised version). While the latter is more related to the rest of the paper, we find the former contribution important to clarify (although it focuses on minimization).
>
> > Some related works are missing. Gidel et al. 2019b presents the negative momentum method, which leads to a similar “backtracking step” in the vector field the gradients. CLC-GAN [Xu et al. arXiv:1909.13188] proposes to stabilize GANs’ training dynamics by interpreting the gradient flow as a dynamic system and manage it with closed-loop control, which is also related to this paper.
>
> Thanks for your recommendation, we added these references.
>
> > The experiments are conducted on 4 representative datasets. However, the image sizes (32*32) do not match the state-of-the-art GANs, which typically can generate larger images (e.g. Mescheder et al. 2018). One concern is whether the proposed method works well on larger datasets such as LSUN and ImageNet with higher resolutions. It would be more convincing if the authors can provide results on larger datasets.
>
> While we don’t have the computation budget to run such experiments*, our extensive benchmark also addresses the case of high sample variability by considering 32x32 ImageNet. While it would be interesting to see large resolution results we consider our findings (over different base-optimizer--see also results in App.) relevant for a large portion of the ICLR audience (who don’t have such resources or use smaller scale setups). Also, see Sec.4 of our last revised version, which provides further insights into why Alg.1 helps for games --which combined with the compelling empirical results increases the chances that the performance gain could follow for larger-resolution setups.
>
> \* By quoting BigGAN's authors "On 8xV100: this script takes 15 days to train to 150k iterations” (referring to 128x128 pixel resolution), we estimate a cost of 15days:  24h x 8 GPUS x 2.5 = 7200$ on Google Cloud for a *single* experiment.

---

### Official Review · AnonReviewer2 · 2020-10-27
**I find that the results on the bilinear game are convincing, but the results on GANs are much less convincing.**

**Rating:** 4
**Confidence:** 4

**Review:**

This paper adapts a recently introduced Lookhead method in optimization to improve the training of minmax problems such as GANs. The main challenge is to address the variance of stochastic gradients and the rotational component in the Jacobian of the gradients. The algorithm proposed in this paper shows improvements over existing methods, in terms of convergence rate. The stability of existing algorithms is a major issue when applying GANs on image dataset, the proposed Lookhead minmax method also improves the stability. I find that the results on the bilinear game are convincing, but the results on GANs are much less convincing.

There seems to me some inconsistency in the numerical results in Tab. 1 and Fig. 6. Are the same hyper-parameters used for LA-Alt-GAN on ImageNet? The median diverges in (c) of Fig. 6 while in Table 1, the FID is around 14.37? This is hard for me to interpret as the total number of iterations are the same, which is 500k.

In terms of writing, I would recommend to focus on the minmax problem. In this sense, the section 3 is a bit distracting as it is about the optimization problem. The Algorithm 5 (page 19) about LA-AltGAN is important to understand the results, so it would be better to explain it more in the main body of the paper. Also what is LA-ExtraGrad algorithm? How Adam is applied in these algorithms? These proposed variants are not very clear to the reader. Also is the number of passes the same as the number of iterations?

To make the results in Fig 5. more conclusive, I would recommend to zoom into the eigenvalues of LA-AltGAN, to see how small the imagery parts of the eigenvalues are. This is an interesting observation that it seems not very conclusive to show that LA-GAN shows no rotations.

Overall, I think both the results and the writing need to be improved.

---

> ### Author Response · Authors · 2020-11-15
> **Thanks for the thorough comments**
>
>
> > There seems to me some inconsistency in the numerical results in Tab. 1 and Fig. 6. Are the same hyper-parameters used for LA-Alt-GAN on ImageNet? The median diverges in (c) of Fig. 6 while in Table 1, the FID is around 14.37? This is hard for me to interpret as the total number of iterations are the same, which is 500k.
>
> Thanks for the question. Table 1 reports the *best* scores ever obtained per method while computing the computationally-expensive IS and FID scores once every few thousands of parameter updates (and then averaging these over 5 seeds)--as reported in all related works that we are aware of [all refs in Table 2]. Note that contrary to classification, GAN training often diverges on real-world applications (also with 5:1 update ratio of D:G, see our paragraph on stability), and GAN methods in practice store a copy of the best performing model obtained during the training and return this model. We updated the caption to make this clearer. Thus, the result of 14.37 is the mean of the best FIDs of the LA-AltGAN methods obtained over 500k iterations (not at the 500K iter).
>
> > In terms of writing, I would recommend to focus on the minmax problem. In this sense, the section 3 is a bit distracting as it is about the optimization problem. The Algorithm 5 (page 19) about LA-AltGAN is important to understand the results, so it would be better to explain it more in the main body of the paper. Also what is LA-ExtraGrad algorithm? How Adam is applied in these algorithms? These proposed variants are not very clear to the reader. Also is the number of passes the same as the number of iterations?
>
> We added explicit descriptions of both LA-EG and LA-GAN-Adam in the appendix (Alg. 4 & 5), and we re-wrote Alg.1 more generally. By using a “looser” pseudo-code of Alg.1 in the paper, we aim to point out the general “meta/wrapper” nature of LA--Minmax: the fact that it can be combined with any "base”-optimizer. Please see our revised beginning of Section 4 and Alg.1. We are considering your recommendation regarding Section 3 and hope to address it in our next revision (before the author response deadline).
> The number of passes is different from the number of iterations, where the former is normalized by the computation and denotes one cycle of forward+backward pass, or alternatively can be expressed as the number of “gradient queries” (this term is taken from the optimization literature, and the SVRE reference). In other words, using parameter updates as the x-axis could be misleading as for example, extragradient uses extra passes (gradient queries) per parameter update. The number of passes is thus a fair indicator of the computational complexity (as wall-clock time measurements can be relatively noisier).  Due to lack of space, we added this description on the “number of passes” in the Appendix (and we reference it from Section 4.1 in the main paper).
>
> > To make the results in Fig 5. more conclusive, I would recommend to zoom into the eigenvalues of LA-AltGAN, to see how small the imagery parts of the eigenvalues are. This is an interesting observation that it seems not very conclusive to show that LA-GAN shows no rotations.
>
> The eigenvalues of LA-AltGAN have zero imaginary part (they are not zoomed out). Interestingly, to our knowledge, this case has not been observed in (Berard et al ‘19)--which potentially opens up directions.

---

> > ### Author Response · Authors · 2020-11-20
> > **Update on Sec.3**
> >
> > Due to the extended page limit (relative to the initial submission), we'd keep Sec. 3 as we consider it an important clarification, and we added extended theoretical discussions concerning games in Sec. 4 (see last revised version).  Moreover, we modified the bullet points for our contributions inline with this modification.
> >
> > Please let us know if your concerns are addressed.

---

> > > ### Comment · AnonReviewer2 · 2020-11-21
> > > **A decent contribution**
> > >
> > > Thanks for the authors for updating the paper, I find the results on GAN  more convincing now. I am happy to raise my score to Accept. Given the instability of training, if the main results could be made reproducible, that would be better.

---

> > > > ### Author Response · Authors · 2020-11-24
> > > > **Our source code is provided**
> > > >
> > > > Thanks a lot, the code is provided [here](https://anonymous.4open.science/r/e062872f-2ff7-4aa8-9c74-59c215098577/) (anonymously for the moment).

---

### Official Review · AnonReviewer3 · 2020-10-27
**This paper treats with algorithmic schemes that improve the training of GANs. These iterative methods are experimentally evaluate in a extensive manner.**

**Rating:** 7
**Confidence:** 2

**Review:**

The authors are focusing in this paper on the development of LookAhead mechanism. Their contributions are the following:

1. They provide an improvement upon an already existing convergence rate for the minimization case.
2. They propose the application of the LookAhead mechanism (under the necessary adjustments) for the more general min/max framework.
3. These results are extensively justified by various numerical experiments.

The paper is well-written and easy to follow. Moreover, the core idea of applying the LookAhead for mix-max problems, as far as my knowledge goes, is  novel. In addition, without being an expert, the experiments seem promising and extensive.
That said, my concern would be regarding the theoretical part of the paper. I fully understand that the paper is primarily experimentally oriented. However some more theoretical analysis would be useful. More precisely, the theoretical justification for the improved convergence part for the minimization part seems to be more like a sketch rather than a rigorous mathematical proof. Overall, I suggest that it would be very interesting to see some provable theoretical guarantees starting with the  paper's motivation example that of the bilinear game.

---

> ### Author Response · Authors · 2020-11-20
> **Thanks for your valuable comments!**
>
> Thanks a lot for your review and recommendations! While we aim to keep this paper primarily empirical as you noted, to address your main concern we improved our Sec. 4 (see last revised version), where, starting from a simple bilinear game we show analytically that: *(i)* GDA diverges on this example, while in contrast *(ii)* Alg.1 combined with GDA can converge for this example for some alpha. We then show that Alg.1 converges locally when combined with a converging base optimizer (Thm. 2 in the revised version).

---

### Author Response · Authors · 2021-04-28
**Source code**

Our code is available here: https://github.com/Chavdarova/LAGAN-Lookahead_Minimax

Thanks to all our reviewers!

---

### Decision · Program_Chairs · 2021-01-07
**Final Decision**

**Decision:**

Accept (Poster)

**Comment:**

The reviews were a bit mixed, with a general consensus towards acceptance. The authors were one of the first to extend lookahead to minimax optimization, and demonstrated its potential through thorough experiments. The theoretical results were not as strong or at least not very well presented. Overall, the authors made interesting contributions and this work is of general interest to the ICLR audience. Please consider further polishing the draft according to the reviewers' comments. The AC would also like to draw the authors' attention to the following issues discovered in an independent assessment:

(a) As the reviewers mentioned, how lookahead-minimax addresses rotational dynamics is not clearly presented. The current justification is a bit handwaving and speculative.

(b) Please consider rewriting Section 3. If there is some new results on the minimization problem, state the results in a theorem and include all assumptions clearly and precisely. This is also useful for other people to reference your result. As the authors themselves pointed out, this result falls quite short of explaining or motivation lookahead.

(c) Theorem 1, add e.g. in the citation before (Bertsekas, 1999). Theorem 2, in its current form, is quite weak in two aspects: (a) without checking its proof one can already see how to derive it in 1 line or 2. (b) if the base optimizer already converges, what is the point of having lookahead to converge as well? The potentially different convergence rate should be one's target here. It is certainly fine for the authors to not fully justify their proposed algorithm, as long as the authors (hopefully) are at least aware of the issues.

(d) Section 4 is a bit disappointing as one would have expected the authors to derive some qualitative results here (also raised by some reviewers).

---

> ### Author Response · Authors · 2021-04-28
> **Final updates**
>
> Thanks for your feedback!
>
> Besides minor polishing, we rewrote section 3 more clearly, and also explained more precisely for the given example how Lookahead handles the rotational dynamics.
>
> While Theorem 2 is simple, it's interesting that it's general -- one does not need to analyze convergence given different converging base optimizers. On the other hand, such analyses for a non-converging base optimizer depend on the hyperparameter selection what is out of the scope of this work.